# Molecular Mechanisms of Neuroimmune Crosstalk in the Pathogenesis of Stroke

**DOI:** 10.3390/ijms22179486

**Published:** 2021-08-31

**Authors:** Yun Hwa Choi, Collin Laaker, Martin Hsu, Peter Cismaru, Matyas Sandor, Zsuzsanna Fabry

**Affiliations:** 1School of Pharmacy, University of Wisconsin-Madison, Madison, WI 53705, USA; ychoi253@wisc.edu; 2Neuroscience Training Program, University of Wisconsin-Madison, Madison, WI 53705, USA; claaker@wisc.edu (C.L.); mhsu24@wisc.edu (M.H.); 3Chemistry, University of Wisconsin-Madison, Madison, WI 53705, USA; pcismaru@wisc.edu; 4Department of Pathology and Laboratory Medicine, University of Wisconsin-Madison, Madison, WI 53705, USA; msandor@wisc.edu

**Keywords:** stroke, lymphatics, lymphangiogenesis, CNS neuroinflammation

## Abstract

Stroke disrupts the homeostatic balance within the brain and is associated with a significant accumulation of necrotic cellular debris, fluid, and peripheral immune cells in the central nervous system (CNS). Additionally, cells, antigens, and other factors exit the brain into the periphery via damaged blood–brain barrier cells, glymphatic transport mechanisms, and lymphatic vessels, which dramatically influence the systemic immune response and lead to complex neuroimmune communication. As a result, the immunological response after stroke is a highly dynamic event that involves communication between multiple organ systems and cell types, with significant consequences on not only the initial stroke tissue injury but long-term recovery in the CNS. In this review, we discuss the complex immunological and physiological interactions that occur after stroke with a focus on how the peripheral immune system and CNS communicate to regulate post-stroke brain homeostasis. First, we discuss the post-stroke immune cascade across different contexts as well as homeostatic regulation within the brain. Then, we focus on the lymphatic vessels surrounding the brain and their ability to coordinate both immune response and fluid homeostasis within the brain after stroke. Finally, we discuss how therapeutic manipulation of peripheral systems may provide new mechanisms to treat stroke injury.

## 1. Introduction

Maintaining homeostasis is important during both steady-state conditions and disease for survival. As every organ system collaborates to accomplish this goal, the lymphatic system is considered a key regulator of fluid drainage, removal of waste products, and transportation of immune cells and lipids [1,2,3]. However, the central nervous system (CNS) has been characterized as an “immune-privileged” region, unlike peripheral organs, because it does not have conventional lymphatic vasculature, and barriers tightly limit the presence of peripheral immune cells [1,4,5,6,7]. The observation of significant brain-infiltrating immune cells in neurodegenerative and neuroinflammatory diseases and the recent re-discovery of lymphatic vessels surrounding the CNS, such as dural meningeal lymphatics, basal lymphatics, and lymphatics near the cribriform plate (CP), has opened new approaches to the treatment of CNS diseases [1,6,8,9,10,11,12].

Stroke is a primary example of neuropathology that has significant homeostatic disruption, immune cell infiltration, and an imminent need for novel therapies [13,14,15]. Even though stroke can result in death and long-term disability with significant complications, there is unfortunately only one FDA-approved drug, recombinant tissue plasminogen activator (rtPA), for ischemic stroke patients [15,16,17,18,19,20]. Due to several adverse effects and limitations involving the timing of administration, rtPA is given to only about 2–6% of ischemic stroke patients [21,22,23,24]. Hemorrhagic stroke patients receive medications to relieve intracranial and blood pressure or to remove anticoagulants [25]. Several attempts to develop therapeutics for stroke patients have failed in clinical trials [26,27,28,29], which exemplifies the need to understand the complicated immunological mechanisms of stroke.

In this review, we briefly discuss dynamic immune responses across different contexts during the stroke and the function of lymphatic vessels surrounding the CNS during this process. We also discuss how lymphatic systems play a role in communicating between CNS and the periphery to mediate homeostasis after stroke.

## 2. Cascades of Immune Responses after Stroke

There are two types of strokes: ischemic and hemorrhagic. Ischemic stroke occurs more commonly (87% of stroke patients) due to an occlusion of blood vessels [30,31,32,33,34]. In humans, the middle cerebral artery is the most common location of occlusion caused by a buildup of plaque in the artery (thrombosis) or plaque that originates from other locations (embolism) [30]. Thus, the same artery is occluded using the filament in the animal model of ischemic stroke, and this surgery is called middle cerebral artery occlusion (MCAO) [35]. Hemorrhagic stroke, including intracerebral hemorrhage and subarachnoid hemorrhage (13% of stroke patients), occurs due to a rupture of a blood vessel [36].

### 2.1. Ischemic Stroke

The immediate effects of ischemic injury are induced by the reduction of cerebral blood flow, oxygen, and nutrients, leading to cell death via necrosis and apoptosis in the ischemic core region. During this process, danger-associated molecular patterns (DAMPs), which include high mobility group box-1 (HMGB1) and peroxiredoxin family proteins, are released to activate brain resident microglial cells by pattern recognition receptors (PRRs), including toll-like receptors (TLRs) and initiate leukocyte transmigration, by secreting interleukin (IL)-1, IL-6, matrix metallopeptidase (MMP)-9, and tumor necrosis factor (TNF)-α [30,37,38,39,40,41]. In the periphery of the ischemic core, called the penumbra, neurons may survive and provoke pathogenic factors, including calcium overload, oxidative stress, and glutamate excitotoxicity, that can cause more cell death [30].

#### 2.1.1. Leukocyte Adherence after Stroke

The sudden change in shear blood flow rates after ischemia increases the expression of selectins, which bind with circulating leukocytes. This adherence of leukocytes further clogs the vessels to contribute to brain injury and prevent reperfusion after ischemic stroke [30]. Thrombin in the blood induces multiple actions such as expressing adhesion molecules in endothelial cells, acting as a chemotaxin for leukocytes, and activating the complement system [30,42,43]. Anaphylatoxins of the complement system, including C3a and C5a, bind to complement receptors on myeloid cells to induce the release of free radicals and secrete pro-inflammatory cytokines [30,44,45]. It has been shown that genetic deletion or treatment with antagonists of the C3a receptor reduced stroke infarct [46]. Additionally, deficiency of mannose-binding lectin, another pattern-binding molecule that activates the complement system, shows improved outcomes after stroke injury in both humans and mice [47,48]. A complement system can induce phagocytosis of damaged neurons in the penumbra by microglia and/or macrophages in the chronic phase [45]. Together, these immediate effects of ischemic injury within the brain establish the foundation of the immunoregulatory environment for later mechanisms of damage, repair, and infiltration of immune cells from the periphery.

Platelets are recruited in the cerebral venules after ischemic and reperfusion injury. The formation of platelet-leukocyte aggregates in the endothelial cells induces cell activation by generating superoxide and platelet-activating factors and mediates inflammatory responses in the body [42]. The activated endothelial cells increase the expression of various adhesion molecules, such as selectins and intercellular adhesion molecule 1 (ICAM-1) [42,49,50,51]. P-selectin is induced as early as 15 min in the ipsilateral cerebral cortex after the injury, while E-selectin and ICAM-1 can be expressed after 1–2 h of insult. All these adhesion molecules persist for several hours after the ischemia [52,53,54], and the elevated level of selectins in plasma can be used as a marker of acute and chronic inflammatory states in ischemic stroke patients [55]. Administering CY-1503, an analog of sialyl-Lewis X (sLe^x^), which is a ligand of selectins, or the anti-E-selectin antibody reduced the number of neutrophils and infarction after MCAO [56,57].

The accumulation of leukocytes, which are mostly neutrophils (about 85%) and non-polymorphonuclear (non-PMN) leukocytes, starts as early as 4 h and increases up to 48 h after reperfusion [42]. The leukocyte rolling mediated between P-selectin and neutrophils is the dominant adhesion mechanism in the early phase of injury, whereas leukocyte adherence via interaction of ICAM-1 and β2-integrins (CD11a/CD18 and CD11b/CD18) on non-PMN leukocytes governs the later phase of injury [42]. P-selectin and ICAM-1 may be involved in platelet accumulation, as a blockade of these adhesion molecules showed a reduction in platelet adhesion through platelet-associated P-selectin glycoprotein ligand 1 (PSGL-1), while inhibition of platelet adhesion glycoproteins GP/IIb/IIIa induced little effect [58,59].

PMN leukocytes can also bind to endothelial cells with the glycoprotein CD11b/CD18 complex. When this interaction was inhibited by blocking CD18 using monoclonal antibody IB4 before reperfusion, reflow was increased in microvessels, capillaries, pre-capillary arterioles, and post-capillary venules [60]. Treatment with a CD11b/CD18 antagonist reduced severity and improved functional output but showed no effects on infarct volume. Combination therapy of this antagonist and rtPA reduced infarct volume, improved functional recovery, and lowered neutrophil accumulation [61].

Additional pathways of leukocyte adhesion are CD40/CD40L and Notch signaling pathways [42]. CD40 is a membrane glycoprotein expressed in lymphocytes, dendritic cells (DCs), platelets, neuronal cells, and endothelial cells. The interaction between CD40 and its ligand, CD40L, during inflammation is critical to produce pro-inflammatory cytokines and induce cellular adhesion molecules. After an ischemic stroke in patients, CD40L is upregulated in platelets, and CD40 expression is increased in monocytes. When MCAO was induced in CD40- or CD40L-deficient mice, reduced infarct volume, less leukocyte and platelet rolling, and decreased permeability were observed [62]. Notch signaling has shown beneficial effects such as neurogenesis and synaptic plasticity, but it can increase leukocyte infiltration and induce neuronal cell death by apoptotic cascades involving caspase-3 and activating microglia after ischemic stroke. Arumugan et al. showed functional improvements and more minor brain damage after ischemic stroke in mice by using the λ-secretase inhibitor, which is an enzyme releasing a Notch intracellular domain (NICD) [63].

However, targeting these adhesion molecules and signaling pathways provides beneficial effects only in transient ischemia, not in permanent ischemia, by mediating both thrombogenic responses and inflammation to improve stroke outcomes in patients [42]. In clinical trials of stroke patients, treatments using either humanized CD11b antibody or Enlimomab which is ICAM-1 monoclonal antibody have not shown beneficial results. This not only shows the complexity of immune responses in humans but also shows they may involve the upregulation of alternative adhesion molecules after treatment. Another possible explanation is that these drugs may be effective only after reperfusion when leukocyte adhesion occurs; however, only a small percentage of patients would undergo complete reperfusion after ischemic stroke [64,65].

#### 2.1.2. Routes of Immune Cells toward the CNS after Stroke

There are several routes that immune cells can take advantage of to enter the stroke lesion within the brain. After stroke onset, the blood–brain barrier (BBB) is opened, as observed by the increased transcytosis and number of endothelial caveolae within a few hours. However, the tight junctions, such as occludin and zonula occludens 1 (ZO-1), are still stable up to 24 h post-MCAO, and tight junction proteins and vascular endothelial cells start to degrade after about 24–48 h when a second opening of the BBB occurs. MMPs can cause reversible damage on tight junction proteins during both BBB openings, while cyclooxygenase 2 (COX-2) affects the BBB during the second opening. When inhibitors of MMP and COX-2 were used, MMP inhibitors showed effects during the early phase and the COX-2 inhibitor showed benefits in the second opening, indicating that the use of a combination of the inhibitors would be more protective against BBB disruption [66]. BBB disruption then leads to the infiltration of peripheral immune cells [66,67]. Neutrophils may enter the ischemic hemisphere through the binding of C-X-C motif chemokine receptor (CXCR)1 and C-C chemokine receptor type (CCR)5 on the neutrophils with chemoattractants, such as C-X-C motif chemokine ligand (CXCL)1, CXCL2, CXCL5, C-C motif chemokine ligand (CCL)2, CCL3, and CCL5. Monocyte recruitment depends on the CCL2/CCR2 interaction. T-cells utilize lymphocyte function-associated antigen (LFA)-1/ICAM-1, very late antigen (VLA)4/vascular cell adhesion molecule (VCAM)-1, and PSGL/P-selectin signaling to interact with endothelial cells. Regulatory T (Treg) cells and γδ T-cells are recruited via CCL5/CCR5 and CCL6/CCR6 mechanisms, while natural killer (NK) cells are recruited through CXCL-10/CXCR3 and C-X3-C motif chemokine ligand 1 (CX3CL1)/C-X3-C motif chemokine receptor 1 (CX3CR1) mechanisms [41].

The choroid plexus can be another gateway for the entry of immune cells. Epithelial cells in the choroid plexus express ICAM-1, VCAM-1, and CCL20, which can interact with IL-17-secreting lymphocytes through CCR6 to enter the cerebrospinal fluid (CSF) [68,69]. Furthermore, monocytes can enter the CSF through CD73 expression by the choroid plexus and when the CD73 is deleted, exacerbated damage was observed after ischemic injury indicating that these monocytes may have neuroprotective roles [70,71,72]. The choroid plexus also contains major histocompatibility complex (MHC) II^+^ CD169^+^ CD68^+^ macrophages and CD11c^+^ DCs that are possibly derived from CCR2^+^ Ly6c^high^ monocytes. These macrophages express strong CX3CR1 levels and secrete IL-1β under peripheral lipopolysaccharide (LPS) attack, while the DCs can present antigens for T-cells in the choroid plexus for activation [73]. 

#### 2.1.3. Immune Cascades after Stroke

After infiltration into the lesion, dynamic immune cascades occur almost simultaneously to induce both detrimental and beneficial effects after stroke onset (Figure 1) and continue in the later phase (Figure 2). This immune system is summarized in Table 1.

##### Microglia

Microglia, the brain-resident immune cells, respond very quickly to enact both beneficial and detrimental roles, which may last for weeks [74]. Neuronal cell death can decrease colony stimulating factor 1 receptor (CSF1R) ligand IL-34 to deplete microglia, but deficiencies of CX3CR1 and CD200 after cell death activate microglia after stroke [75,76,77]. Moreover, microglia express several receptors after ischemic stroke, such as P2X7 receptor, which can regulate ATP signals; triggering receptor expressed on myeloid cell 2 (TREM2) to suppress inflammatory responses and neuronal apoptosis; ST2 (a member of IL-1R family), which can interact with IL-33; and B-cell-activating factor (BAFF) to produce interferon (IFN)-γ and IL-10 [78,79,80,81]. Even though the depletion of microglia can worsen the stroke outcome, microglia can intensify neuroinflammation by upregulating MMPs and secreting pro-inflammatory cytokines such as TNF-α, IL-1β, reactive oxygen species (ROS), reactive nitrogen species (RNS), and inducible nitric oxide synthase (iNOS), which can worsen the outcome after stroke [74,82,83,84]. Another way that microglia influence immune responses is through the manipulation of astrocytes. Astrocytes can increase neutrophil recruitment via CXCL chemokines, express adhesion molecules for leukocyte recruitment, which can be stimulated by IL-1, and further intensify pro-inflammation by producing cytokines such as IL-1α and TNF-α after ischemic and reperfusion injury of stroke [42,85,86]. Microglia can suppress these astrocyte activities by expressing neurotrophic factor insulin-like growth factor 1 (IGF-1), while phagocytose recruits neutrophils at the penumbra [82,87,88]. Treatment with the CSF1R inhibitor resulted in an increased number of neutrophils and lesion size. This indicates that more neutrophils may enter the ischemic injury site with microglia depletion [87].

##### Neutrophils

Neutrophils can enter the brain parenchyma through abluminal sites of leptomeningeal vessels as early as a few hours after onset [89,90]. They can induce damage in the ischemic area by causing neuronal death, destruction of the BBB, and edema by producing elastases, MMP-9, IL-1β, and ROS over 7 days after the injury [91,92,93,94,95,96]. Neutrophil extracellular traps can further activate platelets and thrombotic processes [39]. The detrimental role of cytotoxic neutrophils is caused in part by TLR4 signaling. When TLR4 signaling is deleted, neutrophils show neuroprotective behavior [97]. Depletion of neutrophils using an anti-neutrophil monoclonal antibody (RP3) in rats or P-selectin^−/−^ mice resolved brain edema, improved reflow, and reduced brain infarct size after MCAO [54,98]. Another group used anti-neutrophil serum to develop neutropenia in rats. This decreased brain edema by about 70%. Interestingly, neutrophils did not infiltrate the brain until infarction was formed in the brain [99]. On the other hand, neutrophils can exhibit neuroprotective effects after the injury. The MMP-9 produced by neutrophils can damage the BBB, but it can also degrade DAMP signaling [100]. It has been shown that inhibition of MMP-9 in the later phase can impair vascular remodeling and worsen stroke outcomes [101]. Neutrophils produce vascular endothelial growth factor (VEGF), which promotes cerebral angiogenesis after stroke [102,103]. A subset of neutrophils, N2 neutrophils, secrete anti-inflammatory molecules such as annexin-1, resolvins, and protectins in order to resolve acute inflammation and prevent chronic inflammatory states [104,105]. Resolvins and protectins represent a class of lipid-derived mediators generated from the metabolites of omega-3 fatty acids [106] that have been studied for their potential immunosuppressive role in stroke. Resolvins work by binding to G-protein coupled receptors (GPCRs) expressed on immune cells and the vascular epithelium to initiate a variety of immunoregulatory effects such as reduced neutrophil mobilization, reduced pro-inflammatory cytokine release, and a decrease in blood pressure [107]. When neutrophil membrane-derived nanovesicles loaded with resolvin D2 (RvD2), a subclass of resolvins, were intravascularly delivered into mice after MCAO, there was a reduction of infarct volume and a reduced secretion of TNF-α, IL-6, and IL-1β in the brain [108]. Additionally, it was shown that these anti-inflammatory effects worked through the binding of these resolvin-containing neutrophil-like nanoparticles with vascular endothelial barrier cells and not direct infiltration into CNS [108]. Similarly, neuroprotectin D1 (NPD1), which can also be produced by neutrophils, elicits similar protective effects through transient receptor potential cation 6 (TRPC6) pathways when administered by i.v. in the rat MCAO model, leading to a reduction in infarct volume [31,109]. Since resolvins and protectins are generated through omega-3 fatty acid metabolism, dietary fatty-acid supplementation has been investigated for both stroke prevention and intervention but has generated inconsistent results in human clinical trials [110].

##### Astrocytes

Astrocytes have stellate morphology and express glial fibrillary acidic protein (GFAP), which can be changed under stress and inflammation. In normal conditions, astrocytes mediate ionic systems, produce neurotransmitters, regulate cerebral blood flow, maintain water homeostasis, and modulate immune responses [111]. Astrocytes collaborate with neurons in order to maintain homeostasis. Astrocytes can transport glucose from plasma into the brain via glucose transporter (GLUT)1 on their end feet. The glycolytic astrocytes metabolize glucose by phosphofructokinase and produce lactate from pyruvate through lactate dehydrogenase 5. The lactate is secreted by astrocytes via monocarboxylic acid transporter (MCT) 1 and MCT 4 and then is transported to neurons via MCT2. This absorbed lactate is converted back to pyruvate and enters the tricarboxylic acid (TCA) cycle in neurons to produce ATP. Unlike astrocytes, neurons use oxidative phosphorylation in their mitochondria to produce energy. Since this process may induce the risk of oxidative stress and glutamate excitotoxicity, neurons downregulate glycolysis. This unique glucose regulation pathway is called the astrocyte neuron lactate shuttle (ANLS) hypothesis. Additionally, glutamate is produced from presynaptic terminals at the synapse during neuronal activity, and astrocytes can remove glutamate along with Na^+^ ions via astroglial glutamate transporters. This further stimulates glycolysis within astrocytes to produce more lactate for neurons. Thus, a sustained increase in neuronal activity increases glycolysis and production of lactate as well as the secretion of glutamate. In opposition, Bak et al. argue that lactate is used for the malate–aspartate shuttle because depolarization increases intracellular Ca^2+^ concentration, which ultimately decreases glycolysis in astrocytes. Even though there are still controversies about the concept, the ANLS hypothesis has been widely accepted to explain the relationship between astrocytes and neurons [112,113,114].

Oxidative stress, disruption of ionic balance, and deregulation of signaling pathways after ischemic stroke can overwhelm astrocytes, which leads to transient hyper-glycolysis as astrocytes attempt to restore ionic imbalance and cellular energy [111,115,116]. This produces a high concentration of lactate in the extracellular fluid, which can be observed in stroke patients [112,116,117,118,119]. Additionally, a disrupted ionic gradient induces calcium influx after ischemia. It opens mitochondrial permeability transition pores in the inner membrane and causes the release of cytochrome c through the ruptured outer membrane to activate caspase-dependent apoptosis [112,116]. Astrocytes express PRRs to sense DAMPs after the injury and overexpress GFAP, called astrogliosis, which is observed in mostly lesion areas. Glial scar formation due to changes in tissue structures can inhibit axon regeneration, causing an increase of pro-inflammatory factors and chronic pain. Reactive astrocytes can be detected within 3–5 days after the onset [111]. Astrocytes express MHC II to present antigens to CD41 T-helper cells and to communicate with cytokines and other neurotransmitters [111,120]. In addition to the pro-inflammatory cytokines, astrocytes produce IL-15 to recruit more CD8^+^ T-cells and NK cells and exacerbate tissue injury after MCAO in mice [121]. Astrocytes are resistant to apoptosis by TNF-related apoptosis-inducing ligand (TRAIL) through calcium/calmodulin-dependent protein kinase II [111,120]. Astrocytes can contribute to neuroprotection by using CD38 signaling to transfer mitochondria to neurons after stroke, and inhibition of this signaling showed worse outcomes [122]. Astrocytes can release several vasoactive substances, including nitric oxide, ATP, and COX, as well as vascular growth factors and glucose to maintain the BBB [111]. Astrocytes can produce transforming growth factor beta (TGF-β), glial-derived neurotrophic factor (GDNF), angiopoietin 1 and 2 (Ang1 and Ang2), VEGF, and basic fibroblast growth factor (bFGF) to stimulate the development of blood vessels and endothelial progenitor cells [111,123].

##### Mast Cells

The majority of mast cells contain granules and reside in the meninges in the CNS [124]. Mast cells can be recruited into cerebral tissues via the GLUT1 isoform on blood vessels after about 24 h of stroke onset [125,126]. Then, mast cells increase vascular permeability and the accumulation of PMN leukocytes through the disrupted BBB and the secretion of histamine, heparin, other vasoactive agents, and proteases in brain tissues during the early stage after stroke. Mast cells may damage the BBB by releasing a large amount of granules, such as vasodilatory mediators, proteases, and histamines, simultaneously. As a result of the destructed BBB, histamine and its receptor can be used to promote cerebral edema after injury [124]. Additionally, histamine and protease tryptase can loosen the vascular–endothelial–cadherin-mediated adhesion of endothelial cells [127]. Additionally, chymase produced by mast cells can degrade MMP-inhibiting proteins such as tissue inhibitor of metalloproteinases (TIMP)-1, cleave fibronectin, and activate procollagenases [124]. Mast cells can activate pro-MMP-2 and pro-MMP-9 to release MMP-2 and 9, respectively, which can further destroy tight junction proteins such as occludin and claudin [124,128]. The hemorrhagic formation after ischemic stroke may be induced by heparin (a strong anticoagulant) and disrupt hemostasis [124]. Animals that received mast cell inhibitor (cromoglycate) or a mast cell-degranulating drug (compound 48/80) showed similar cerebral blood flow. However, brain edema and BBB leakage were reduced after cromoglycate, but both were increased after compound 48/80. The cromoglycate-treated group showed less infiltration of neutrophils than the control group. Mast cell-deficient rats showed a reduction of cerebral edema, BBB disruption, and neutrophil infiltration compared to the control group [129]. Promisingly, mast-cell-deficient rats and cromoglycate-treated rats that were additionally treated with rtPA after ischemic stroke showed decreased brain edema, BBB leakage, and reduced hemorrhagic formation and mortality [124,129,130].

##### Monocyte-Derived Macrophages

Monocyte-derived macrophages (MDMs) enter the ischemic area as early as 3 h after the onset via CCL2-CCR2 signaling [131,132]. The expression of CCL2 has been identified in the cortex, mainly on endothelial cells and astrocytes after MCAO [133,134,135]. CCL2 deficiency in mice showed protection after permanent MCAO by limiting monocyte recruitment and reducing IL-1β protection [136]. However, CCR2^−/−^ mice restrict the entry of MDMs and are associated with reduced TGF-β1 and collagen-4 expression in the brain, which leads to vascular instability and hemorrhagic transformation. CCR2 inhibitors or anti-CCR2 antibodies (MC-21) showed similar results, including increased inflammation and hemorrhagic transformation, and inhibited long-term functional recovery by reducing anti-inflammatory genes [132,137]. It indicates that when the CCL2-CCR2-mediated entry of MDMs to the injury sites is inhibited, the stroke outcome worsens [131,132]. Two subsets of monocytes were observed in mouse blood: CCR2^+^ Ly6C^high^ inflammatory monocytes during acute inflammation and CX3CR1^+^ Ly6C^low^ monocytes patrolling in the repairing process. After three days post-MCAO in mice, there was an accumulation of CCR2^+^ monocyte/macrophages, not CX3CR1^+^ cells, in the ischemic area. After 14 days, CX3CR1^+^ cells were observed at the infarct border zone. CX3CR1^+^ cells also showed three different phenotypes over time, while CCR2^+^ cells showed amoeboid morphology only. CX3CR1^+^ Ly6C^low^ cells were not observed in CCR2-deficient mice, indicating CX3CR1^+^ Ly6C^low^ cells may be differentiated from CCR2^+^ Ly6C^high^ cells instead of being recruited from the blood. However, the exact mechanisms are still unclear [138]. MDMs showed peak infiltration at day 3 after MCAO in mice. At day 7 post-MCAO, MDMs exhibited both pro- and anti-inflammatory phenotypes, which later became mostly anti-inflammatory after the subsequent 2 weeks [132]. This could be influenced by the expression of CD206 on the MDMs, which is associated with protective effects by 1–5 days post-injury but starts to decrease after day 7 post-injury [139,140]. After MDMs become anti-inflammatory, IL-4 and IL-13 signals activate transcription factors, including signal transducer and activator of transcription 6 (STAT6) and peroxisome proliferator-activated receptor gamma (PPARγ), to secrete IL-10, TGF-β, CD302, CD163, platelet-derived growth factor (PDGF), fibronectin 1, and arginase 1 for tissue repair and wound healing [141,142]. Once in the brain, MDMs exacerbate tissue damage and inflammation by releasing pro-inflammatory cytokines [143]. Depleting the macrophages prior to the stroke showed less BBB disruption and improved functional outcome, even though there were no differences in the infarct volume [144]. CCL20, mainly expressed from astrocytes, interacts with CCR6 on microglia to recruit macrophages and produce pro-inflammatory factors, including TNF-α and IL-1β [145]. IL-4-induced macrophages may participate in promoting anti-inflammatory and functional recovery following stroke as well [143]. Administration of IL-33 induced macrophages to produce Th2 cytokines such as IL-4, which is important to trigger long-term functional recovery [146].

##### Dendritic Cells (DCs)

DCs are the professional antigen-presenting cells (APCs) that express MHC II to promote T-cell activation [143]. Migratory DCs can travel from tissues to lymph nodes via the interaction of CCR7 with CCL19 and CCL20 [147]. Additional pathways such as S1P signaling or MHC invariant chain (CD74) can be used for DC migration [148,149]. When those DCs arrive in the lymph nodes, the antigens can be transferred to lymph node-resident DCs to activate T-cell priming via either connexin 43 gap junctions from macrophages to CD103^+^ DCs or exosomes that contain MHC II-enriched compartments with CD86 [150,151]. However, DCs are usually not found in the brain parenchyma [143]. On the contrary, OX62^+^ DCs have been found in dura mater, leptomeninges, and choroid plexus in rats, where there are more opportunities to encounter CNS antigens [152]. After inducing transient MCAO in mice, DCs accumulate in the ischemic hemisphere at 24 h. When these DCs were further specified, the DCs in the core area of the infarct were from the periphery, whereas resident DCs mostly occupied the penumbra [153]. A subtype of conventional DCs, Xcr1^−^ CD172^+^ cDC2, is known as a major population contributing to damage in the mouse ischemic brain. These cDCs secrete IL-23 to increase IL-17 secretion from γδ T-cells, which increases CXCL1 expression on astrocytes to increase neutrophil infiltration [154]. In the permanent MCAO model in rats, Kostulas et al. observed elevated levels of OX62^+^ DCs at 1 h in the ischemic hemisphere, which also showed a correlation with increased brain lesion areas over time. The OX62^+^, OX6^+^ DCs expressed MHC II and stayed elevated at 6 days in the ischemic hemisphere. Several cytokines that may induce pro- or anti-inflammatory signals were expressed in the OX62^+^ DCs, such as IL-1β, IL-6, IL-12, IFN-γ, IL-17, TNF-α, and IL-10, in the ischemic hemisphere [155]. All CD11c^+^ DCs in a mouse model of photothrombosis expressed MHC II^+^, CD40^−^, CD80^+^, and CD86^+^, indicating an important interaction between immature DCs and T-cells in the lesion and subcortical areas [153,156]. However, the role of DCs is not clearly described in humans. In acute ischemic stroke and acute hemorrhagic stroke patients, significantly reduced numbers of both circulating myeloid DC precursors (mDCPs) and plasmacytoid DC precursors (pDCPs) were observed, which recovered after a few days. This transient decrease of circulating DCPs may be a result of their recruitment from the blood into the ischemic brain. The patients with lower DCPs show bigger stroke lesion sizes and co-localization of myeloid DCs with T-cells around cerebral vessels in the infarcted area, indicating that antigen-mediated T-cell activation and long-lasting immune responses have occurred in the brain [157].

##### Natural Killer (NK) Cells

NK cells are CD3^−^ innate lymphocytes that can infiltrate the lesion and border zone of the lesion in stroke patients [158]. In mice after transient MCAO, NK cells were in the peri-infarct areas as early as 3 h after MCAO and remained elevated for at least 4 days after the onset [159]. However, NK cells were still detected after 30 days of MCAO. CX3CR1-expressing NK cells are recruited to the ischemic brain via the interaction of CX3CR1 and CX3CL1, which are overexpressed by injured neurons. NK cells can develop infarction by secreting IFN-γ in T- and B-cell-independent mechanisms and stimulate local inflammation by secreting pro-inflammatory cytokines such as IFN-γ, IL-17a, TNF-α, IL-1β, IL-6, IL-12, and ROS after MCAO. When anti-NK1.1 monoclonal antibodies were administered prior to MCAO in mice, the infarct area was smaller and neurological deficits were less severe, confirming a detrimental role of NK cells [158].

##### T-Cells

As T-cells are associated with inducing leukocyte adherence and thrombo-inflammation, reducing T-cell infiltration can be protective after the injury [160]. Rag-1^−/−^ or severe combined immunodeficient (SCID) mice showed protection after MCAO, further supporting the detrimental roles of T-lymphocytes after stroke [42,160]. Early T-cell activity after onset can be detected within 1–3 days and may not depend on either antigen recognition or T-cell receptor co-stimulation [161,162]. Instead, this may involve angiotensin II in the brain after ischemic injury since the brain injury was alleviated with an angiotensin receptor inhibitor in wild-type mice [163]. Angiotensin II induces its damaging activities via an angiotensin receptor, AT-1, which activates T-cells to release TNF-α without antigenic stimulus [42,163]. Rag-1^−/−^ mice do not develop deleterious effects in response to angiotensin II administration, whereas overexpression of angiotensin genes in mice has shown worse output [164,165]. Other T-cells undergo antigen-dependent mechanisms to be activated after ischemic stroke. Ischemia increases MHC I antigens in the brain, which activates CD8^+^ T-cells. The activated CD8^+^ T-cells secrete cytotoxic molecules, including perforin and Fas ligand, to induce cellular damage and apoptosis. Additionally, IFN-γ in the brain released post-stroke can increase MHC II expression in endothelial cells and astrocytes to prime CD4^+^ T-cells [42]. Among the CD4^+^ T-cells, Th1 and Th17 cells produce pro-inflammatory cytokines such as IL-2, IL-12, TNF-α, IL-17, IL-21, IL-22, IFN-γ, and IL-23 while anti-inflammatory cytokines such as IL-4, IL-5, IL-6, IL-10, and IL-13, can be released by Th2 cells [42,166,167]. Some studies have shown that cytotoxic CD8^+^ T-cells are recruited as early as 3 h and stay for about 30 days, while CD4^+^ T-cells are observed after 24 h and last for about 30 days [168,169]. In a transient MCAO model in mice, T-cells showed detrimental effects in brain injury [160]. Studies have shown that there is an increased expression of IL-23 by infiltrated macrophages and DCs on the first day post-injury, and this affects γδ T-cells to produce IL-17a within 3 days, which synergistically increases CXCL1 expression on astrocytes to promote the infiltration of neutrophils. Analysis of mice that are genetically IL-17- and IL-23-depleted showed IL-23 affects immediate ischemic/reperfusion injury, while IL-17 has delayed responses in the penumbra [86,154,170]. The increased expression of these γδ T-cells contributed to brain infarction in both mice and patients after the stroke. Depletion of γδ T-cells reduced ischemia and reperfusion injury [170]. Interestingly, IL-17a expression was upregulated by reactive astrocytes in the later phase, which improved brain tissue repair and recovery [171]. Another cytokine of interest is IL-21 which is produced by CD4^+^ T-cells and contributes to reperfusion injury after transient MCAO [172].

##### Regulatory T-Cells (Tregs)

In later periods, Tregs are recruited via CCR5 and expanded locally via IL-2, IL-33, and serotonin signaling to exhibit neuroprotective abilities by producing IL-10 [173,174,175]. Treg cells derived from IL-10-deficient mice did not show protective effects against the injury, emphasizing the important role of IL-10 in mediating Treg cells [42]. Tregs can suppress astrogliosis, regulate neurotoxicity of astrocytes, and induce functional recovery [174]. However, Kleinshnitz et al. showed a detrimental side of Treg cells. The group used genetically modified depletion of Treg cell (DEREG) mice, in which FoxP3-expressing cells can produce both green fluorescent protein (GFP) and diphtheria toxin receptors. From this, FoxP3-expressing Treg cells can be selectively depleted using the diphtheria toxin. After inducing transient MCAO in mice, they confirmed a reduction of brain infarct and functional improvement until day 4 in Treg-depleted DEREG mice compared to control DEREG mice. Similar results were observed when Treg cells were depleted after MCAO. Adoptive transfer of CD4^+^ CD25^+^ Treg cells before MCAO in wild-type mice induced bigger stroke infarct size at day 1 post-MCAO. Interestingly, adoptive transfer of CD4^+^ CD25^+^ FoxP3^+^ Treg cells from DEREG mice into Rag1^−/−^ mice also induced infarct in the brain at day 1, independent of IL-17, IFN-γ, and CCR6. When CD4^+^ CD25^+^ Treg cells were reconstituted into Rag1^−/−^ mice after treating the mice with platelet-depleting serum or LFA-1 inhibitor that inhibit interaction with ICAM-1, the damaging effects were overcome by showing smaller brain infarct. This indicates that Treg cells may produce brain injury by interacting with activated endothelial cells and platelets in the early phase of MCAO [176]. A study showed that a large number of CD4^+^ CCR6^+^ FoxP3^+^ Treg cells are found in blood and lymphoid tissues in humans that produce IL-17 after activation and inhibit CD4^+^ T-cell proliferation. Moreover, in certain conditions when T-cell receptors are activated by IL-1β, IL-2, IL-21, and IL-23 in serum, CD4^+^ CCR6^−^ FoxP3^+^ Treg cells differentiate into IL-17-producing cells [177]. In stroke patients, activated T-cells have been observed within 60 days after the onset. Among T-cells, increased expression of CD4^+^ CD25^+^ FoxP3^+^ Treg cells and CD4^+^ CD28^−^ T-cells are noticed in stroke patients. The earlier Treg cells show neuroprotective effects, whereas the latter T-cell population can be used as a predictive marker of poor outcome [42].

##### B-Cells

B-cells can also contribute to neuroprotection after the ischemic stroke, depending on IL-10 [178,179]. Unlike T-cells, B-cells did not show improvements against inflammation and brain injury when MCAO was induced in B-cell-deficient mice [160]. B-cell-deficient μMT^−/−^ mice showed larger infarcts and a higher mortality rate after 48 h of MCAO compared to wild-type mice. After 48 h, accumulation of leukocytes such as neutrophils, T-cells, microglia, and macrophages that produced IFN-γ and TNF-α was observed in the ischemic hemisphere of μMT^−/−^ mice. Adoptive transfer of IL-10^−/−^ B cells to μMT^−/−^ mice did not show a reduction in infarct volume after MCAO, showing that B-cells can restore the beneficial outcome, including inhibition of pro-inflammatory cytokine release from T-cells after stroke through IL-10 production [178]. Similarly, adoptive transfer of naïve B-cells from wild-type mice to MCAO-induced mice generated smaller infarcts at 3 and 7 days. The same experiment with IL-10-deficient B-cells did not show similar protection. Using whole-brain volumetric serial two-photon tomography, diapedesis of B-cells was observed in distant areas from the injury area, such as the cerebral cortex, dentate gyrus, olfactory areas, and hypothalamus, where motor and cognitive abilities are governed, to promote long-term recovery. In line with these observations, the depletion of B-cells impacted post-stroke neurogenesis and cognitive functions [179]. Another study showed that B-cells showed delayed infiltration in the lesion at around 7 weeks after the onset. During that period, B-cells were closely associated with T-cells and CD11c-expressing cells, potentially DCs, for antigen transfer to induce isotype switching. Some of the CD19^+^ B cells co-expressed CD138, indicating these cells are plasma cells. After 7 weeks of stroke in mice, an increased level of IgG (IgG1 and IgG2b), IgA, and IgM was observed in the lesion area. B-cell-deficient μMT^−/−^ mice did not show IgG in the lesion area and cognitive deficit after stroke even though the infarct size and T-cell infiltration were similar to wild-type mice. This indicates that cognitive ability after stroke is mediated by the antibodies produced by B-cells. When mice were treated with CD20 antibodies to ablate B-cells, it also prevented cognitive deficits and IgG expression after stroke [180]. However, Schuhmann et al. showed that depleting B-cells pharmacologically using anti-CD20 24 h before the MCAO in mice or using B-cell-deficient JHD^−/−^ mice that do not have circulating B-cells did not affect the stroke lesion size, number of neutrophils and monocytes, and levels of TNF-α and IL-1β at day 1 and 3 post-MCAO [181].

### 2.2. Hemorrhagic Stroke

A hemorrhagic stroke happens less frequently than ischemic stroke but has a high morbidity and mortality rate (about 40%) [182]. It is caused by the rupture of the cerebral artery wall, leading to blood leakage in either the subarachnoid space, called subarachnoid hemorrhage (SAH), or the tissues of the brain and ventricles, called intracerebral hemorrhage (ICH) [183]. ICH can also happen after the ischemic stroke and is associated with hematoma expansion, edema, and intraventricular hemorrhage [184,185,186].

The outcome of SAH is considered to be associated with the expression of MMP-9, TNF-α, and IL-6, while IL-10 can be related to early brain injury and risk of pneumonia [187]. C-reactive protein (CRP) is also correlated with severity and death at three months in SAH [188]. Another biomarker of SAH is circulating cell-free DNA (cfDNA). The cfDNA is released from necrotic cells and has been recognized recently as a biomarker for cancer, trauma, and ischemic stroke [189,190]. SAH patients showed worse outcomes when the expression of mitochondrial cfDNA was increased [191]. The progression of hemorrhage can be chronic because it can continuously trigger neurologic deficits over an extended period [25]. 

Recently, it has been shown that immune cascades are very similar in ischemic and hemorrhagic stroke [157]. Hemorrhagic stroke can initiate inflammatory responses, induce cerebral edema, and disrupt BBB with neurotoxic materials such as thrombin, fibrin, and erythrocytes [183]. The accumulation of erythrocytes can initiate inflammation by activating TLR4 in the microglia to release TNF-α and IL-1β [192]. It was observed that HMGB1 secreted by monocytes and macrophages interacts with TLR2, TLR4, and TLR5 to upregulate nuclear factor kappa-light-chain-enhancer of activated B-cells (NF-kB). Hence, the expression level of HMGB1 in plasma and CSF can be related to SAH outcomes [193,194]. As the inflammation proceeds, leukocytes are recruited via adhesion molecules to secrete diverse chemokines and cytokines to induce endothelial cell death, recruit more immune cells, and damage the tight junctions in the BBB [183]. Microglia and macrophages phagocytose erythrocytes and degraded debris after the injury [195]. T-cells can be detrimental, but Tregs can be neuroprotective in a rat model of hemorrhagic stroke, as mentioned for ischemic stroke [196].

## 3. Brain Edema and the Glymphatic System

Since the brain parenchyma does not have lymphatic vessels, waste products and cellular debris are thought to be cleared through the glymphatic pathway [197]. The CSF flows from the brain parenchyma to subarachnoid spaces through periarterial and perivenous spaces, depending on convective fluid movement, which can be driven by cerebral arterial pulsation [198,199,200]. During this process, small molecules are exchanged between the CSF and interstitial fluid (ISF) [200]. After a stroke, slower fluid drainage and waste removal have been shown, which is possibly due to reactive astrogliosis induced by cytokines released from necrotic cells [201,202,203]. Astrogliosis causes the dispersion of aquaporin (AQP)4 in the cytoplasm of brain parenchyma [204,205], leading to fluid accumulation in the brain parenchyma, tissue swelling (cerebral edema), cell death, and increased intracranial pressure (ICP) [206,207]. This significantly inhibited CSF flow can be observed up to 30 days even after the resolution of ICP, indicating that cerebral lymphatic drainage and homeostasis is significantly disrupted after stroke [208].

There are two types of brain edema after stroke [206]. Cytotoxic edema can happen with insufficient ATP production after injury and the accumulation of intracellular sodium ions with the malfunction of the Na^+^/K^+^ pump. Vasogenic edema occurs due to the disruption of the BBB, oxidative stress with ROS, or reperfusion, which can introduce excessive fluid in the brain and cause the swelling of astrocytes [66,209,210].

### 3.1. Aquaporins (AQPs)

AQPs are small, hydrophobic intrinsic plasma membrane proteins expressed in several cell types that mainly transport water bidirectionally in order to facilitate fluid and solute transport in various osmotic conditions. Within the AQP family, AQPs can be categorized into AQPs (AQP1, 2, 4, 5, 6, and 8), which are strictly for water, and aquaglyceroporins (AQP3, 7, and 9), which can transport water and other small solutes, including glycerol [211,212,213,214]. Two electrode voltage-clamp measurements showed that AQPs can exclude ions and protons during transport [212]. Additionally, the asparagine–proline–alanine (NPA) region and aromatic-arginine (ar/R) motif can be utilized as filters for selectivity. The NPA region is located in the middle of the AQP pores, which function as the main barrier of the proton. The ar/R motif is located toward the extracellular side of the pore and determines the selectivity of permeable solutes between AQPs and aquaglyceroporins. The AQP pore size may be one of the factors affecting solute permeability, but the location of the specific residues in those filters affects solute exclusion and specificity [213]. 

Recently, AQPs have been found to transport not only water but other neutral polar solutes passively along the osmotic gradient, increasing the cell permeability of those molecules [213,215]. AQP9 is mostly expressed in hepatocytes and transport purines, pyrimidines, and carbamides, along with water [216]. AQP1 can transport carbon dioxide gas. When CO_2_ transport by AQP1 was inhibited in oocytes, water transport was also inhibited, which was not observed with the AQP1 mutant [217,218].

### 3.2. AQPs in Stroke

Among those AQPs, AQP4 is abundantly expressed in astrocytes in the CNS and mediates water transport across the BBB and blood–spinal cord barrier (BSCB) [211,219]. While glial-conditional knock-out mice showed about 31% reduction in cerebral edema, AQP4 knock-out mice showed reduced post-ischemic brain edema by about 35%. This indicates that AQP4 plays critical roles in cytotoxic edema [219,220,221]. Kitchen and colleagues showed that the AQP4 expression level increases in response to hypoxia. Acute hypoxia localized AQP4 in cortical astrocytes and increased water permeability. AQP4 localization depends on calmodulin (CaM), which binds to the carboxyl terminal of AQP4 and changes its conformation. Inhibition of the CaM using trifluoperazine (TFP) in a rat model of spinal cord injury prevented AQP4 localization, reduced edema, and increased functional recovery [219]. Similar results were observed in the photothrombotic stroke model in mice. TFP reduced brain edema by inhibiting the expression of AQP4 without inducing changes in electrolytes and other metabolic markers in the brain [211]. However, depleting AQP4 expression reduces glymphatic flow and encourages more protein deposits in mice [200]. It was shown that AQP4 can induce cytotoxic edema in the early stage but removes fluid during vasogenic edema in the later period [211,219]. It has also been suggested that glymphatic system dysfunction may be involved in the pathogenesis of post-stroke dementia after studying different types of neuroinflammation models, such as stroke and traumatic brain injury (TBI) in mice [197,222,223].

At 1 and 48 h after MCAO, AQP4 expression increases (which correlates with cerebral edema) to develop protective mechanisms [224]. IL-1 and HMGB1 could contribute to the upregulation of this AQP4 expression [225]. AQP4 knock-out mice showed a reduction of osmotically driven water transport, which induced less ischemic volume, less neuronal apoptosis, and higher survival rates, while overexpression of AQP4 exacerbated cerebral edema and acute water intoxication [226,227]. The beneficial effect of AQP4 deficiency after ischemic stroke in mice was shown when cytotoxic edema was the predominant pathophysiological mechanism. This could be also possible due to the lower expression of Ca^2+^ signaling, reduced Cl^−^ current, removal of K^+^ through connexin 43, and decreased glutamate uptake with the lower expression of AQP4. During vasogenic edema, AQP4 deficiency showed more edema and increased ICP, indicating the elimination of water into CSF and blood is dependent on AQP4 [224]. The excessive fluid may be cleared by bulk flow into CSF or through the glial lymphatic system [210,228].

AQP1 is expressed in epithelial cells of the choroid plexus. As the choroid plexus produces CSF mainly driven by an osmotic gradient with Cl^−^/HCO3^−^ and Na^+^/H^+^, the AQP1 also shows high osmotic water permeability. The AQP1 is co-localized with growth-associated protein 43 (GAP-43), hinting that it may be associated with neuroplasticity and repair mechanisms [224].

AQP9 is also induced around peri-infarct areas, mainly in the cortex, after about 24 h of MCAO in mice, possibly participating in water homeostasis and leukocyte transmigration through an increased number of filopodia. The AQP9 permeability to lactate would be useful to buffer lactate levels after lactic acidosis during ischemic stroke [229,230].

However, several studies have raised questions to better understand the glymphatic system. Some researchers consider that the glymphatic system may be limited to only the fluid movement via AQP4 because the spaces are not large enough for immune cells and macromolecules [200,231,232]. Mestre et al. further studied this to show that spread of depolarization and subsequent vasoconstriction created larger spaces in perivascular channels that CSF could utilize to enter the brain within a few minutes after MCAO in mice [233]. Another suggested pathway of CSF is using the extracellular spaces in capillaries and arteries to exit the brain in the opposite direction, towards arterial blood flow [234]. Alshuhri and colleagues recently pointed out the limitation of studying the glymphatic system using labeled tracers (700–938 Da), which are larger than water (18 Da). Hence, the group used water molecules as a tracer instead to study the glymphatic hypothesis and AQP4. H_2_^17^O (19 Da) or gadolinium-diethylenetriamine pentaacetate (Gd-DTPA; 938 Da), as control, was injected into CSF at the cisterna magna in rats. Serial MRI images showed the rapid penetration of H_2_^17^O into the parenchyma of entire brain regions, not capillaries. Unlike the control, H_2_^17^O did not accumulate in the subarachnoid spaces and ventricles. The group assumed both the diffusion and bulk flow of ISF would contribute to the distribution of H_2_^17^O since it was a very rapid process. The penetration of H_2_^17^O was reduced with an AQP4 inhibitor (TGN-020) [235].

TGN-020 has been suggested as one of the potential candidates of AQP4 inhibitors. Huber et al. showed its inhibitory effects on AQP4-mediated water transport in a dose-dependent manner in an osmotic swelling assay using *Xenopus laevis* oocytes. The group also used modeling software to identify that TGN-020 showed the most stabilized docking energies in the AQP4 protein monomer [236]. However, the osmotic swelling assay used in this experiment was measured once per minute. Since oocyte volume can be affected by several factors such as ion transporters, the physical properties of oocytes, cell viability, and regulatory mechanisms, it is recommended to record it every second or better to calculate more accurate initial oocyte swelling rates and curves [237]. Additionally, the group could only model a single protein monomer of the entire AQP4, which is a protein homotetramer, due to insufficient gating mechanisms and a lack of information on the protein binding sites [236]. This could neglect the critical lipid interactions of AQP4 in their modeling computations [237]. Thus, several potential AQP4 inhibitors suggested by Huber et al. have been challenged on their efficacies [237,238].

However, a single intraperitoneal injection of TGN-020 before MCAO in mice reduced cerebral edema and cortical infarction, as confirmed with MRI images 24 h post-MCAO [239]. Similar results were observed when a single dose of TGN-020 was given after MCAO in rats by showing a reduction of cerebral edema, gliosis, and apoptosis at 3 and 7 days post-MCAO [240]. When *Xenopus laevis* oocytes expressing rat AQP4 were treated with TGN-020, AQP4-dependent swelling and water permeability were reduced, with an IC_50_ of 3.5 μM. Additionally, TGN-020 showed selectivity toward AQP4 when the inhibitor was added to oocytes expressing other AQPs. TGN-020 showed similar efficiency in the inhibition of mouse AQP4 and human AQP4 in oocytes [241]. TGN-020 did not affect the stimulus-evoked K^+^ transient, indicating that AQP4 is not required for K^+^ dynamics [241,242]. Moreover, TGN-020 administered both pre- and post-spinal cord injury showed better functional recovery and reduced edema, with the inhibition of glial scar formation and less tissue destruction [243,244]. Even though TGN-020 shows promising results after ischemic stroke and spinal cord injury, TGN-020 also has an affinity to AQP1, which is similar to AQP4, with about 60% homology [245]. Hence, it is still necessary to be cautious in analyzing in vivo data involving TGN-020 treatment.

Since it is not clear what the roles and functions of the glymphatic system are, more research into both healthy and disease states during CNS neuroinflammation will be necessary.

## 4. Lymphatic System

Lymphatic vessels absorb CSF that is produced from the choroid plexus and exit the skull via the CP, jugular foramen, and internal carotid artery to cervical lymph nodes (CLNs) [8,246,247]. Since 1869, when lymphatic drainage was first described, several studies have shown that the CP is the main route for CSF to reach the cervical lymphatic system [248,249,250,251]. Later in the 1960s, initial and collecting lymphatics in the dura mater were described, and their connection with cervical lymphatic systems was characterized [247,252]. CSF can also be drained through lymphatics at the base of the skull [247,253]. Thus, there are several routes of CSF drainage in the CNS, but their relative contribution to stroke is still unknown [246].

Additionally, the re-discovery of lymphatic endothelial cells (LECs) in the leptomeninges has ignited interest in the area and shown the diverse role the lymphatic system plays around the CNS. Most importantly, these lymphatics could be important routes to facilitate communication between the CNS and the periphery by draining CSF and its components [248,254,255,256].

### 4.1. General Structure and Roles

Conventional lymphatic vessels are present in nearly all organs in vertebrates. The main functions are the removal of ISF from blood capillaries and the maintenance of immunosurveillance in tissues [257]. Most LECs that are lined in lymphatic vessels are differentiated from venous progenitors during development [258]. Transcription factors SRY-Box transcription factor 18 (Sox18) and chicken ovalbumin upstream promotor transcription factor II (Coup-TFII) induce lymphatic regulator prospero homeobox 1 (Prox1), which migrates from the cardinal vein through VEGF-C and vascular endothelial growth factor receptor (VEGFR)-3 signaling [259]. All the LECs express several markers such as Prox1, lymphatic vessel endothelial hyaluronan receptor (lyve)-1, VEGFR-3, and podoplanin [257]. Prox1 is a nuclear transcription factor that is the major controller, and it is highly expressed in both lymphatic capillaries, pre-collectors, and secondary valves in the collecting lymphatics. However, the expression of Prox1 is lower in the collecting lymphatics. Lyve-1 is an integral membrane glycoprotein that is highly expressed in lymphatic capillaries [260,261]. VEGFR-3, a tyrosine kinase receptor, is more expressed in lymphatic capillaries as it manages lymphangiogenesis by interacting with both VEGF-C and VEGF-D [260,262]. VEGF-C can be produced from various types of sources, including smooth muscle cells, macrophages, fibroblasts, and even tumor cells [257,260,263]. Podoplanin is a mucin-type transmembrane glycoprotein that is expressed throughout lymphatic vessels [260,264]. Other additional signaling pathways govern lymphatic development and mechanisms such as adrenomedullin (Adm)/calcitonin receptor-like receptor (CLR)/receptor activity-modifying protein (RAMP) 2 [265], hyaluronan [266,267], collagen- and calcium-binding epidermal growth factor (EGF) domain 1 (CCBE1)/a disintegrin and metalloproteinase (ADAM) metallopeptidase with thrombospondin type 1 motif 3 (ADAMTS3) [268,269], Ang-Tie [270,271], and forkhead box C2 (FOXC2) [272,273].

The lymph fluid is transported from capillaries to the collecting vessels, which ultimately lead to lymph nodes and systemic blood circulation. Highly branched lymphatic capillaries have thin basement membranes and oak-leaf-shaped endothelial cells. This shape functions to create one-way valves that facilitate the formation of the lymph from ISF [260]. Lymphatic capillary cells are connected by discontinuous junctions (buttons) that can increase permeability and act as the entrance point for ISF and immune cells [260,274]. Pre-collectors are lymphatic vessels with a single layer of endothelial cells and secondary valves [275]. These vessels can work as canals in between capillaries and collecting vessels [274]. Collecting vessels also have basement membranes, connected by continuous cell–cell junctions (zippers), and are surrounded by a single, continuous layer of smooth muscle cells [260,274]. The contraction of these muscle cells and arterial pulsations drives lymph propulsion. The intraluminal lymphatic valve is assembled with endothelial cells and connective tissues. These valves prevent the backflow of the lymph and draw borders between lymphatic capillaries and collecting vessels [260].

LECs can also present antigens to CD8^+^ T-cells and stimulate T-cell proliferation [276]. In mice, LECs in lymph nodes can archive antigens to present them to migratory DCs directly or through LEC apoptosis [277]. Then, DCs present the antigens to T-cells. As a result, LECs can have important roles in both immune tolerance and memory [278].

### 4.2. Meningeal Lymphatic Vessels

Meningeal lymphatics were first described by Mascagni and colleagues, then Louveau and Aspelund re-discovered and characterized these lymphatic vessels using modern methodology [6,8]. Meningeal lymphatic vessels (mLVs) are developed postnatally with VEGF-C that is produced by blood vascular smooth muscle cells. The blockade of VEGF-C interaction with VEGFR-3 during development impairs mLV formation. However, mLVs undergo lymphangiogenesis with excess VEGF-C, indicating mLVs have regenerative plasticity [263].

Dural mLVs can drain CSF and ISF, which has been observed in rodents [6,8,279], humans, and primates [280]. During this process, macromolecules and immune cells can drain into CLNs to maintain immune surveillance in the brain. Dural mLVs do not have valves and smooth muscle cells around them. However, they have button-like noncontinuous junctions that have similar phenotypical characteristics to peripheral lymphatic capillaries. Additionally, dural mLVs express markers of LECs including Prox1, VEGFR-3, Podoplanin, lyve-1, and CCL21 [6,8].

Basal mLVs have a mixture of button-like noncontinuous and zipper-like junctional patterns, so they reflect both lymphatic capillary and collecting lymphatic vessel characteristics. These lymphatic vessels do not have smooth muscle cells around them but do have valves. Thus, they can uptake fluid and maintain the unidirectional flow [8,9,263]. Basal mLVs are closely located to subarachnoid space, so there are spatial advantages in their ability to absorb and drain CSF and its macromolecules. These functions are reduced with aging [9].

The mLVs facilitate CNS-lymph node crosstalk. Stroke patients show more myelin expression and increased association of neural antigens with lymph node macrophages near the activated T-cells [281]. DCs recognize these brain antigens in the CSF and upregulate CCR7 expression to interact with the CCL21 expressed by mLVs, which facilitates DC migration toward CLNs [282,283]. In CCR7-deficient mice, both DCs and T-cells could not drain into the CLNs [254]. After DCs arrive in the CLNs, they initiate antigen-specific T-cell and B-cell activation and proliferation. Activated T-cells and antibodies are transported to the blood vessels and enter the brain to exert their pro- or anti-inflammatory effects [284]. One type of innate immune cells found in meninges after ischemic brain injury is γδ T-cells. These cells can produce IL-17, which is critical after ischemic injury in the brain [86,170]. After SAH, mLVs drain the extravasated erythrocytes from subarachnoid space into CLNs [285].

If the drainage routes become dysfunctional due to mLV ligation, the accumulation of antigens and other pro-inflammatory factors can occur, which affects cognitive ability, including learning and memory [286,287]. In the mouse model of Alzheimer’s disease, mLVs after photoablation failed to clear meningeal amyloid-β [286]. However, the blockade of mLVs in experimental autoimmune encephalomyelitis (EAE), a mouse model of multiple sclerosis, showed a reduction in disease severity and the alleviation of inflammatory responses by less activating CCR7^+^ T-cells in lymph nodes [254].

### 4.3. Lymphatics near the Cribriform Plate (CP Lymphatics)

The CP is a porous ethmoid bone plate that separates cranial and nasal cavities. The foramina allow CSF to flow through intercellular spaces in olfactory nerves, lyve-1^+^ vessels, and blood vessels [288]. Lymphatic vessels penetrate the CP and are closely attached to perineural cells and fibroblasts of the olfactory nerves to prevent CSF leakage into the interstitium [289,290]. In rats, there is no arachnoid barrier layer near the CP as it is in the meninges [291]. Walter et al. described that the CP has pocket-like areas in subarachnoid spaces [248]. Additionally, AQP1 was observed to be associated with the olfactory bulbs, nerve junctions, and foramina of the CP. This was further supported by observing the absence of arachnoid villi in fetuses and its presence in adults, even though the CSF formation rates are similar [292,293,294,295].

There is significant evidence that CSF drains through the CP in humans and other mammals [248,249,250,279,296,297,298,299,300]. Several studies have injected tracers or GFP^+^ CD4^+^ T-cells to identify how CSF drains along the olfactory nerves through the CP to deep cervical lymph nodes (dCLNs) [296,301,302]. Johnston and colleagues injected Microfil into subarachnoid compartments in multiple species and observed a drainage pathway to the CP [249,303]. India ink was injected into CSF in the human filled perineural spaces of the olfactory nerves, which then appeared in the nasal submucosal tissue [298]. In humans, the CSF flow was confirmed in close proximity to the CP after 3–6 h in all cases using magnetic resonance imaging (MRI) with a tracer. The tracer was also observed in the grey matter of the straight gyrus and deep white matter of the frontal lobe [304]. Another group also confirmed this using dynamic positron emission tomography (PET) in human subjects [250]. When CP lymphatics were occluded, CSF drainage was reduced while ICP was increased [305,306]. The lymphatics near the CP can be a route to connect to the peripheral lymphatic network [1].

Albumin labeled with fluorescent tracer was injected into cisterna magna, and multiple drainage sites were analyzed simultaneously in mice. Results indicated that the CP near the olfactory area and spinal regions were the main CSF outflow sites. CSF elimination rates were greater in the CP region compared to spinal regions in mice. Even though the CSF outflow and elimination rates were reduced in the nasal and CP areas with aging, a similar decline was not observed in the spinal regions. Ex vivo images confirmed that CSF could be drained into subarachnoid space from cisterna magna as well. However, there was no tracer in meninges in both dorsal and basal, optic nerves, and trigeminal nerves [246]. Interestingly, nasal-instilled aqueous solutions were observed in the subarachnoid spaces and meninges, showing that fluid can be exchanged bidirectionally through the CP. However, the ablation of nasal-associated lymphoid tissue (NALT) did not increase IL-17 and showed no effects on brain injury [307].

## 5. Lymphangiogenesis

As there is increasing evidence suggesting that lymphatic vessels play critical roles in maintaining homeostasis, it has been suggested that cerebral lymphatic drainage is important in understanding pathological conditions after stroke [308,309,310]. Disruption of the BBB and accumulation of immune cells after stroke can cause the disruption of lymphatic transport as well as the reduction of CSF flow, potentially leading to brain edema [311].

Lymphangiogenesis is the process of new lymphatic vessel formation from pre-existing LECs through vessel splitting (intussusception), endothelial cell migration, or endothelial cell re-connection [312,313]. Although relative contributions are not clearly known, LEC sprouting is the most described mechanism of lymphangiogenesis. Fine filopodia are formed on the lymphatic capillaries. Then, these sprouts start to proliferate and extend to become mature lymphatic vessels through VEGFR-3 [314,315]. During lymphangiogenesis, VEGF-C undergoes two proteolytic processes that require extracellular protein CCBE1, Adamts3, and other proteases [316,317]. VEGF-C and VEGF-D bind to VEGFR-3, but they can also bind to neuropilin-2 (Nrp2), which is a transmembrane receptor guiding developmental axons. Nrp2 is a co-receptor for VEGFR-3 and mediates lymphatic sprouting [318]. It was observed that VEGFR-3^+^ cells migrate to the tips of sprouting cells and induce lymphangiogenesis, if there was an inadequate postnatal deletion of *Vegfr3* [319].

Cerebral ischemia also stimulates LEC proliferation with VEGF-C in CLNs (Figure 3). Inhibition of the interaction between VEGF-C and VEGFR-3 using the VEGFR-3 inhibitor reduced brain infarct volume and pro-inflammatory macrophage numbers. When superficial CLNs (sCLNs) were removed, the number of neutrophils and macrophages in the brain as well as in the circulation was decreased [320]. However, inhibition of lymphatic drainage by removing both sCLNs and dCLNs exacerbated ischemic brain infarct after MCAO in rats by increasing edema and the levels of sodium and calcium in the ischemic hemisphere [321]. 

One study showed that superior sagittal sinus on mLVs showed lymphangiogenesis after 14 days of photothrombosis in mice, whereas it was not shown in MCAO-induced mice. Additionally, they found *Vegfr3*^wt/mut^ mice showed fewer mLVs and demonstrated larger ischemic volume as well as worse prognosis after transient MCAO [322]. In EAE, lymphatics in the CP underwent lymphangiogenesis, whereas mLVs did not [1]. However, when VEGFR-3 signaling was inhibited, mLVs showed regression while lymphatics near the CP were unaffected [1,254,263].

Macrophages and DCs are frequently found to be associated with the proliferation and permeability of the lymphatic system [323,324,325,326]. Neutrophils can also participate in the lymphangiogenesis process by producing VEGF-C and VEGF-D, participating in VEGF-C proteolytic steps, or interacting with macrophages. Intranodal lymphangiogenesis is dependent on neutrophils when B-cells are absent. However, depleting neutrophils did not cause changes in VEGF-C expression, which shows that neutrophils are not the only source of VEGF-C [327]. Some of the microRNAs (miR) are involved in lymphangiogenesis. miR-466 can bind to Prox1 to reduce lymphangiogenesis [328], and miR-181a can inhibit Prox1 expression directly to reprogram LECs to become blood vascular endothelial cells (BECs) [329]. miR-31 showed direct effects by repressing genes that are specific to LECs, such as FOXC2, and affecting lymphatic development [330]. Both miR-31 and miR-181a can participate in bone morphogenetic protein 2 (BMP2) signaling to regulate LEC identity negatively [331].

Our lab has shown that lymphatics near the CP play significant roles in the drainage of CSF, CNS-derived antigens, and immune cells during EAE. The CP lymphatics underwent lymphangiogenesis during EAE, driven by an interaction of VEGFR-3 and VEGF-C, which were produced by CD11b^+^ macrophages and CD11b^+^, CD11c^+^ DCs. However, treatment with the VEGFR-3 inhibitor after EAE caused the regression of dural mLVs, while lymphatics near the CP remained unaffected after EAE [1]. This suggests that lymphatic vessels in the CNS have heterogeneous roles and responses to VEGF-C/VEGFR-3 signaling during steady-state and neuroinflammation. Different developmental maturities and requirements for sustained VEGF-C maintenance may be one explanation for the observed differences in treatment response, with dural mLVs forming much later in development when compared to other regions like the lymphatics near the CP [263].

## 6. Aging

Stroke is a disease that predominantly affects older people [34]. The risk of stroke in humans increases once people reach the age of 55. In ischemic stroke, prevalence is about 13% for 60–79-year-old individuals and becomes 27% for individuals who are older than 80 years old. The risk of hemorrhagic stroke doubles with each decade [36].

### 6.1. Inflammaging

The hallmarks of brain aging are dysregulated energy metabolism in mitochondria, activation of the inflammasome, oxidative damage, defective autophagy, inflammation, and impaired waste disposal mechanisms. These changes can be uncontrollable during disease [332,333]. Dysfunction of the mitochondria and endoplasmic reticulum can affect cellular Ca^2+^ processing capability, which can increase ROS production and neuronal apoptosis [334,335,336]. Aged smooth muscle cells express a high level of iNOS, which can produce subsequent oxidative stress [337]. The decreased glutamate transporters and glutamate-aspartate transporters reduce the efficient regulation of glutamate concentration for neuronal excitotoxicity [338]. The capacity of phagocytosis is declined in the aged animals, leading to the accumulation of debris such as tau proteins involved in DNA methylation and acetylation, as well as lipid metabolism. Unfortunately, endogenous DNA repair function also decreases with aging. This can cause the functional impairment of neurons and, eventually, neuronal cell death [339,340,341]. During aging, the accumulation of endogenous factors from dysfunctional cells and pro-inflammatory cytokines initiates the innate immune system, especially macrophages, to mediate stress. This persistent presence of stress and immune responses during aging is called inflammaging. Since healthy individuals also undergo the aging process, the plasma levels of TGF-β and cortisol are used as indicators of healthy aging [332,333,342]. When the gene expression of inflammation-related genes in human hippocampal brain tissues was studied, old study participants (60–99 years old) overexpressed those genes compared to young participants (20–59 years old) [343].

With aging, the number of microglia increases, but their functions decrease as they lose flexibility in mobility and phagocytic ability [338]. On the other hand, microglia with aging are polarized to become pro-inflammatory, where they express increased levels of pro-inflammatory molecules but are less sensitive to regulatory signals [344,345]. Aged oligodendrocytes can exhibit swelling morphology, which is the result of degenerated myelin sheaths in the cytoplasm. Under normal conditions, this would stimulate remyelination and recovery steps, but instead, with aging, it goes through irreversible demyelination and leaves incompletely repaired white matter injury [338,345,346]. Aged astrocytes express more GFAP, which can suppress the supportive roles [347]. Morphology of astrocytes in aging brains is changed to a short phenotype with high expression levels of IL-1b, IL-18, NF-kB, and pro-caspase-1 [332]. The level of CD4^+^ CD25^+^ Tregs is increased, but these cells express lower inhibitory function and cytokine production in aged mice [348]. With aging, loss of perivascular polarization of AQP4 and increased BBB permeability are observed; they can affect cognitive decline and give poor functional outcomes after stroke [349,350]. Together, the effects of inflammaging contribute to a changing neuroinflammatory microenvironment across an organism’s life span and can greatly influence stroke outcomes.

### 6.2. Sex and Aging

Stroke shows higher incidence rates in males, and this dimorphic epidemiology presents in children as well [351]. In young and middle-aged mice, females showed delayed development of stroke lesions with less BBB permeability. However, aged female mice showed more ischemic damage than males [352]. In vitro studies have shown that neurons in the cortical plate or ventricular zone display more protein kinase B (Akt) expression in females than neurons in males, which show higher sensitivity to glutamate and peroxynitrate [353,354]. Additionally, astrocytes in females were more resistant to oxygen and glucose deprivation (OGD) via the P450 aromatase. However, more astrocytes in females underwent cell death when OGD was combined with inflammatory mediators [355,356].

Sex hormones such as estrogen and testosterone fluctuate across an organism’s life span. Animal models have shown that estrogen can be beneficial after ischemic stroke but loses its protective roles in aged female animals [357]. Estrogen can downregulate the expression of the major tight junctions, ZO-1, on LECs and make macrophages express an anti-atherosclerotic phenotype [358,359]. Estrogen contributes to phosphorylation of extracellular signal-regulated kinase (ERK) and Akt in endothelial cells, activation of Adm, RAMP2 and RAMP3, and stimulation of survival signaling pathways [360,361,362]. Additionally, estrogen may decrease oxidative stress and activate vasodilation by producing more nitric oxide (NO) from endothelial cells [363,364]. During this step, estradiol (E2) can participate in endothelial nitric oxide synthase (eNOS) production by phosphorylating proto-oncogene tyrosine-protein kinase Src (c-Src) [364]. Even though estrogen seems to serve a protective role, the administration of estrogen in postmenopausal females failed to show this similar protection by unexpectedly increasing the risk of stroke [365].

The amount of E2 is different in males and females, even though they express the same amount of estrogen receptor α (ERα) [366]. ERα interacts with Prox1 in the lymphatic vessels to inhibit associated genes, while E2 may affect the transcription of lymphatic genes via lyve-1, VEGF-D, VEGFR-3, and hyaluronic acid synthases [367]. Deletion of ERα can cause defects in the lymphatic system and lymphatic gene expression [368]. Since diseases related to lymphatics tend to be more common in females and females have more endogenous serum VEGF-C and VEGF-D, further studies are necessary to understand how estrogen and the ER impact VEGF-C expression in lymphatics in different disease models [369,370,371].

### 6.3. Lymphatic Vessels

Aging affects the structures and functions of lymphatic vessels [9,263,279,286]. After the age of 65, the number of collecting lymphatic vessels decreases, and connections between lymphatics are also reduced [372]. Aged lymphatic vessels show less muscle cell induction, with discontinuous and irregular organization [372]. This loss can affect lymphatic valve gating and electrical synapses [372,373]. In aged rats, there was a decrease in muscle contractile proteins, cytoskeleton-associated proteins, and myosin-binding proteins in the lymphatic collectors [374]. Na^+^, K^+^, Ca^2+^ channels involved in muscle cell action potential and cell depolarization are decreased as well and inhibit lymphatic pump activity [374]. Decreased eNOS and increased iNOS activities in aged rats lead to loss of ability to regulate flow and pumping ability [375,376,377]. In older rats, there was no change in contraction frequency and no inhibition of lymphatic pumping ability with high flow [375]. This shows that aging may alter the self-regulatory mechanisms of lymphatic vessels to the flow [376].

LECs are covered by glycocalyx on the lumen side, which functions as a barrier between lymph and endothelium to inhibit the adherence of immune cells and pathogens. In aged lymphatic vessels, this glycocalyx is impaired, and pathogens can enter the surrounding tissues more easily [374]. Additionally, aging affects the adherence of junctional proteins, such as vascular endothelial (VE)-cadherin and β-catenin, to induce hyperpermeability [377,378]. Furthermore, drainage of lymphatic vessels declines with age in both rodents and humans due to decreased lymphatic capillary density and transport capacity of collecting lymphatic vessels [374,379,380,381]. This functional decline has been shown in the thoracic duct, skin, mLVs, and mesenteric lymphatic vessels [9,279,286,375,382]. Aged collecting lymphatic vessels have shown increased permeability with enlargement and decreased contractility, which is associated with oxidative stress [374,379,383,384]. Potentially, the alterations of lymphatic function induced by aging can lead to the development of autoimmunity and the decline of protective immune functions [385].

Recently, it has been identified that coverage and drainage of mLVs were reduced in aged mice, which may be related to cognitive decline [286]. CCR7 expression is also decreased in the meningeal lymphatics with aging. The loss of CCR7 expression leads to diminished IFN-γ-producing CD4^+^ T-cells, effector T-cells, and cognitive ability but increased meningeal Treg responses. When anti-CD25 antibodies were given to aged mice, cognitive function was improved, with a reduction of Treg responses in meninges and dCLNs [386].

## 7. Communication between the CNS and the Peripheral Immune System

There has been an increasing number of studies showing the interaction between the CNS and the periphery after stroke and other CNS neurodegenerative diseases [11,387,388,389]. From an immunological perspective, CLNs and spleen have been focused in this section to discuss potential mechanisms and complications in the periphery after stroke.

### 7.1. Cervical Lymph Nodes and Spleen

Brain antigens from damaged cells, such as enolase, S100b, and GFAP, in stroke patients and mice are drained into lymphoid tissues, lymph nodes, and spleen through CSF and serum, depending on the severity of the injury [281,390,391,392,393]. Signals from the brain can promote systemic inflammation through VEGFR-3 [320]. Brain antigen, a subset of neutrophils, and DCs migrate to CLNs via the interaction of CCR7 and CCL21, with lymphatic vessels after stroke to induce T-cell activation and proliferation, increase autoimmunity, and promote tolerogenic immunity with Treg expression [281,394,395,396,397]. However, ligation or blocking of the dCLNs further exacerbated the deficiency of the glial lymphatic system and induced secondary cerebral ischemia and edema [398,399].

The leaked brain antigens drain into the spleen and activate systemic immune responses, including B- and T-cells [281,390,391]. The activated immune cells disseminate inflammatory factors in the periphery, which may travel to the brain through the disrupted BBB [400,401]. Activated splenocytes after MCAO produced a large amount of pro-inflammatory factors such as TNF-α, IFN-γ, and IL-6 at 6 h and anti-inflammatory factor IL-10 at 22 h through T-cell receptors. Activated T-cells in the lymph nodes and blood in MCAO mice also released pro-inflammatory cytokines and IL-10 at 22 h [402]. IL-6 that was secreted from the spleen remained increased in plasma until about 24 h after MCAO, while a peak expression of IL-6 in the brain was at 24 h coincidingly [403]. IL-6 expression in blood peaked at day 4 in ischemic stroke patients but at day 10 in hemorrhagic stroke patients [404]. Similar to IL-6, CXCL1 showed increased expression in brain, plasma, and periphery tissues such as lung and liver after 4 h in MCAO mice. This trend was decreased at about 24 h when its expression in the brain peaked [403]. In order to further confirm the entry of periphery cells to the brain, the splenocytes were labeled with carboxyfluorescein succinimidyl ester (CFSE). The CFSE+ cells in the spleen were reduced after 48 h of MCAO, while the CFSE-labeled lymphocytes, monocytes, and neutrophils were detected in the blood after 48 h. The CFSE-labeled monocytes and NK cells were found in the brain after 48 h, and T-cells were found in the brain after 96 h of MCAO in mice [405].

In mice, ischemic stroke affects a subpopulation of macrophages in the spleen. The density of both tissue-resident and bone-marrow-derived splenic macrophages increases, which is associated with local proliferation instead of cellular recruitment. Additionally, increased expression levels of lysosomal associated membrane protein 2 (LAMP2) and decreased MHC II expression were observed, indicating increased phagocytosis but less antigen presentation, respectively [406]. Splenectomy in animal models of ischemic stroke, ICH, and TBI showed protective effects [394]. However, these effects are likely not associated with long-term protection after ischemic stroke as delayed splenectomy failed to show protective effects [407].

#### 7.1.1. Peripheral Immunodepression (Lymphopenia)

In general, stroke patients show peripheral immunodepression and lymphopenia, including monocyte deactivation [408]. Reduced levels of splenic T-cells and NK cells and reduced T-cell responsiveness are observed during the stroke-induced peripheral immunodepression [409,410]. Bone marrow-derived suppressor cells were proliferated by HMGB1 from the injured brain to inhibit adaptive immune responses, induce lymphopenia, and stimulate monocyte exhaustion [411]. In humans, levels of catecholamines and cortisol are increased [412,413,414]. Catecholamines inhibit antigen presentation by β2-adrenoceptors, while corticosteroids inhibit APCs to secrete inflammatory cytokines but promote the development of tolerogenic APCs [415,416]. Thus, stroke-induced immunodepression shows tolerogenic immune responses.

Neurotransmitters secreted by the sympathetic nervous system may be involved in this peripheral lymphopenia process in both animals and humans as 98% of splenic innervation are sympathetic nerve fibers [283,410,417]. In fact, rapid activation of the sympathetic nervous system/hypothalamus–pituitary–adrenal (SNS/HPA) axis after stroke is considered the first communication of the immune system between the CNS and periphery [418]. Stroke-induced lymphopenia can increase the activation of the adrenergic system, Th2 cytokine production, and atrophy of the lymphoid organs [410,419]. The administration of the propranolol β-adrenoreceptor blocker reduced mortality by reversing splenic contraction and producing more IL-10 from invariant natural killer T-cells (iNKTs) [420]. Pharmacological inhibition of SNS after ischemic stroke in mice showed less infarct volume, higher long-term survival, and higher CNS antigen-specific Th1 responses [421]. Activation of the SNS increased the number of Treg cells in the bone marrow [422].

The parasympathetic nervous system (PNS) is not directly associated with the spleen but can affect immune responses through vagus nerves and the α7 nicotinic acetylcholine receptor. The activation of PNS can inhibit systemic and neuroinflammation in animal models of stroke [423,424], but the PNS does not directly interfere with splenic contraction [413]. In humans, PNS deactivation showed poor outcomes after hemorrhagic stroke [187].

Illanes et al. studied immune cell composition and cytokine profiles after three different ICH sizes (10, 30, 50 μL). Large ICH showed a reduction of splenocytes, leukocytes, and lymphocytes but a higher number of monocytes in blood and spleen [425]. There is also evidence that the spleen may play protective roles in ICH. When human neural stem cells were infused via an intravenous injection after ICH in rats, upregulated TNF-α, IL-6, and NF-kB levels were attenuated in the brain and spleen. This result was not shown after a splenectomy before ICH [426]. In humans, larger ICH sizes have been associated with splenic shrinkage, lower lymphocytes and NK cells, and increased infection risks [413]. Splenic shrinkage was also observed after stroke [400,413,427]. Splenic atrophy may happen after 48 h of stroke but re-expand to normal after 7–10 days [388,428]. Post-stroke splenic contraction may be related to an increased number of circulating neutrophils [427,428] and pro-inflammatory cytokines [400].

#### 7.1.2. Systemic Inflammatory Response Syndrome

Patients with severe stroke show systemic inflammatory response syndrome. After patients are treated for thrombolysis, they can still develop this syndrome and usually show poor functional outcomes [429]. It is not known whether systemic immunodeficiency syndrome happens in ICH and SAH. One SAH study in human shows that patients with higher clinical severity showed more suppression of CD4^+^, CD8^+^, and NK cells [430]. In ICH, sympathetic stimulation is increased, and increased risks of infection, which indirectly shows SNS-associated immunosuppression, also occurs after ICH [431,432,433].

### 7.2. Complications in the Periphery

Stroke-associated infection (SAI) is the most frequent complication (13–45%), which commonly involves the respiratory and urinary tracts [434,435,436]. Infectious complications also happen after ICH (30% of the patients) and are predictive factors of hospital mortality and readmission rate [437,438]. Interestingly, regardless of the stroke types, ischemic stroke, ICH, and SAH showed an association with urinary tract and pneumonia complications [437,439,440,441]. In fact, up to 65% of ischemic stroke patients and 26–41% of SAH patients experience these complications [439,440,441]. It is also critical to understand the timing of the infections [442]. Bronchial or dental infections can induce atherosclerosis and cause stroke [443]. The previous infection among patients can increase the risk of death in young patients and increase platelet–leukocyte aggregation [444,445].

The reasons and mechanisms of post-stroke infections are still unclear. When permanent MCAO was induced in rats, about 55.5%, 45.4%, and 30% bacterial proliferation were observed in the MCAO group after 24, 48, and 72 h, respectively, whereas bacterial proliferation was not observed in the sham group [446]. The IFN-γ deficiency is observed with bacterial infections, which can be prevented by inhibiting SNS or administering propranolol after ischemic stroke in mice. Adoptive transfer of NK cells and T-cells from wild-type mice or IFN-γ treatment after day 1 of ischemic stroke in mice showed a reduction of bacteria [447]. Additionally, the MCAO group showed higher ileum tissue injury scores than the sham group over time. This study shows that stroke can induce mucosal damage and bacterial translocation. However, the detailed mechanism and source of the infections are not yet clear [446].

#### 7.2.1. Autoimmunity after Stroke

Several studies have shown that inflammation can induce autoimmunity after stroke in animals and humans [448,449,450,451]. SAI can be a source of autoimmunity because it can increase APC maturation and participate in the development of autoimmunity [452,453,454]. Moreover, SAI in stroke patients increased stress hormone levels, increased anti-inflammatory cytokines [448,455,456,457], reduced numbers of circulating lymphocytes, deactivated monocytes, and reduced the production of inflammatory cytokines [455,457,458]. Administering recombinant T-cell receptor ligands (RTLs) that express MHC II in mice after MCAO, showed a reduction of brain lesion size, less accumulation of inflammatory cells, including macrophages and DCs, and decreased splenic atrophy. However, control RTLs without peptides or mismatched MHC II showed no effect on brain infarct size. This study indicates that autoreactive T-cells are antigen-mediated and worsen brain injury [459]. Autoreactive CD4^+^ and CD8^+^ T-cells, as well as B-cells, were increased 4 days after stroke [460]. However, timing can be an important factor in autoimmune responses. Some studies have shown that both CD4^+^ and CD8^+^ T-cells are recruited within 24 h of stroke [168,461], whereas other studies have shown a similar activity happens after 4 days of stroke [162,173,462]. It has been observed that T-cells are present in the human brain after 3–4 days of the stroke, and no autoreactive T-cells were observed in the periphery before 4 days of stroke [460,463]. Hence, it is assumed that the earlier T-cell activity is antigen-independent, and antigen-dependent T-cell activities will occur later in the ischemic tissues [408]. In order to control the autoimmunity, there are regulatory mechanisms, such as clonal deletion, to ensure tolerance in stroke [173,464].

#### 7.2.2. Complication in Gastrointestinal System after Stroke 

Up to 50% of stroke patients experience complications involving the gastrointestinal system, such as microbial dysbiosis and intestinal bleeding [465]. β-adrenergic signaling disrupts production of intestinal mucin, gut permeability and composition of microbes after stroke in mice and upregulates the TREM1 level in intestinal macrophages [466,467,468]. Inhibition of β-adrenergic receptors in mice improved survival rates by lowering bacteremia [410,420]. Interestingly, stimulating the vagus nerve or activating the α7-nicotinic acetylcholine receptor exhibited protective effects but increased pulmonary infections [469,470]. Recently, it was observed that lymphocytes and γδ T-cells can travel from the small intestine to the brain and meninges, which can worsen the injury after stroke [467,471]. During this process, Treg cells in the gut may contribute to the penetration of γδ T-cells into the ischemic brain [471]. This could be an effort to minimize brain damage by limiting the detrimental immune responses.

However, it is still challenging to fully understand the full effects of stroke on systemic inflammation because many studies have been done on animals without comorbidities [472].

## 8. Novel Treatments Targeting Neuroimmune Communications and Homeostasis in Stroke

There have been several attempts to treat stroke through the targeting of the immune system, specifically modulating early components of the immune cascade into the brain. For example, natalizumab, a drug commonly used to inhibit leukocyte infiltration in multiple sclerosis, showed promise in stroke animal models, but clinical trials revealed little success in treating human stroke patients [473,474,475]. In these trials, natalizumab treatment was administered at the early acute stages of stroke, 9–24 h after stroke onset, but it offered no additional protective effects compared to the placebo control. Corresponding mouse anti-CD49d antibodies showed protective effects after cerebral ischemia in mice [476]. Fingolimod reduces T-cell responsiveness through CCR7-mediated lymphocyte retention in lymph nodes [477]. This drug showed treatment effects after ICH [478,479], but controversial results after cerebral ischemia [480,481,482,483]. Statins showed beneficial effects by impairing DCs maturation by lowering MHC II expression. This diminished the antigen presentation ability of DCs to T-cells and further increased the number of tolerogenic DCs and Treg cells [484,485,486]. Another interesting approach is vaccination with oxidized low-density lipoprotein, certain bacteria, or heat shock protein 65 to prevent atherosclerosis [487,488,489].

Recently, targeting spleen and systemic immunomodulation with stem cells has been an attractive therapeutic approach after stroke. There are two cell therapy strategies: the nerve repair strategy (injecting cells to the injury site) and the immunomodulatory strategy. Using various types of stem cells, such as human umbilical cord blood cells, hematopoietic stem cells, bone marrow stem cells, neural stem cells, and human amnion epithelial cells, showed beneficial effects after stroke in animal models [394]. Intracerebrally injecting human bone marrow mesenchymal stromal cells into rats after stroke showed that those cells could migrate to the spleen via lymphatic vessels. The highest number was found after 3 days of injection in both the brain and spleen. The density of inflammatory cells was much higher in the spleen compared to the brain [490].

As discussed earlier, lymphatic vessels are important facilitators of neuroimmune interactions but can be used as an important drug delivery route in cell therapy. Lymphatics can uptake 10–80 nm in diameter particles if they are subcutaneously injected [491]. Various types of drug delivery methods can be considered, including the extracellular vesicles, which are about 30–100 nm [492]. Extracellular vesicles play critical roles in several physiological processes and immune regulation. These vesicles can be involved in inflammatory immune suppression or stimulation, autoimmunity, and infectious diseases [492]. Exosomes have been an attractive gene therapy target because they can carry genetic materials and be transported to lymph nodes through lymphatic vessels [389,492]. Exosomes derived from pluripotent mesenchymal stem cells showed favorable results in the ICH model in rats [493]. Since exosomes can carry genetic information, these can be used as valuable cell-to-cell communicators [394].

The AQP channels of the glymphatic system have also been investigated as potential drug targets in the treatment of stroke. Data from animal models suggest that AQP4 inhibition in astrocytes might be used to reduce brain edema early in ischemic stroke [494]. However, this is complicated by the low identification and validation of AQP modulators, including AQP4 inhibitors, as well as the widespread expression of the 13 homologous AQP water channels across the body, suggesting that specific AQP targeting within the brain will likely be accompanied by significant off-target effects [495,496]. With that said, AQP inhibitors such as TGN-020, which have effects upon AQP4, still offer promise but have never been tested in humans, and more research is needed to validate the efficacy of AQP inhibitors [239]. In addition, it should be noted that broad AQP4 expression in astrocytes is not functionally binary, and variations in membrane localization, channel permeability, and function can exist with no change in the measured AQP4 protein levels [213,497]. Therefore, more nuanced targeting of AQP4 cell surface mislocalization could provide an additional therapeutic pathway to treat post-stroke edema outside broad inhibition.

Finally, while traditional MCAO animal models have provided much of the foundational understanding of the immune response following stroke and potential molecular targets, the next era of stroke research and therapeutic targeting will likely be aided by the application of novel model systems and new screening technologies for drug discovery. Specifically, 3D human brain organoids, in certain contexts, may provide a more translationally relevant system to study stroke-related factors on human cell populations within an isolated system. While traditional brain organoid systems are limited by their lack of vasculature and, most notably, immune cells, it has been demonstrated that brain organoids can still induce a diverse repertoire of brain components, including the hypothalamus, brain stem, and even a CSF-producing choroid plexus with functioning epithelial barrier cells [498,499,500]. Here, stroke-relevant conditions such as deoxygenation and glucose deprivation can be applied in combination with co-cultured immune cells to study the impacts on structure, function, and expressional changes in the system [501]. Additionally, recent advances have allowed for the creation of in vitro vascularized brain organoids with functional components that mirror many major characteristics of the blood–brain barrier, and, interestingly, brain organoids have been shown to be implantable in mice, leading to vascularization by the host in vivo [502,503]. One study has even documented transplanted and vascularized brain organoids in a mouse MCAO model and the post-stroke therapeutic effects of the transplanted organoids, highlighting the organoids themselves as a potential long-term intervention to reduce stroke damage and improve recovery [504]. Microfluidic systems or “on-a-chip” human brain organoids and blood–brain barrier models, in combination with advanced imaging techniques, will be essential tools to fully understand the mechanisms of BBB permeability, endothelial cell activation, leukocyte adherence, and infiltration during stroke-related conditions [505,506,507]. So while the creation of reliable human brain model systems has obvious limitations, there is great promise that as these in vitro systems improve and scale upwards, they can be adapted to high-throughput screening platforms and provide stroke researchers a mechanism to screen large numbers of potential drug targets in parallel [508,509,510].

## 9. Conclusions

It is widely accepted that the consequences of a stroke are rarely contained within the CNS and are instead characterized by both the influx and efflux of cytokines, cells, and fluid that can significantly impact stroke outcomes. As a result, understanding how the CNS and the periphery engage in this bi-directional communication can provide a mechanism to control brain homeostasis and neuroinflammation during stroke. Recent research has indicated that the lymphatic vessels surrounding the brain and the astrocytes within the brain may provide a novel therapeutic interface to modulate stroke-related disruption of fluid homeostasis and influence the coordination of the peripheral immune response. Therefore, future research is needed to fully understand the functional role of lymphatic vessels during stroke across pathological contexts and the positive and negative effects of their manipulation.

## Figures and Tables

**Figure 1 ijms-22-09486-f001:**
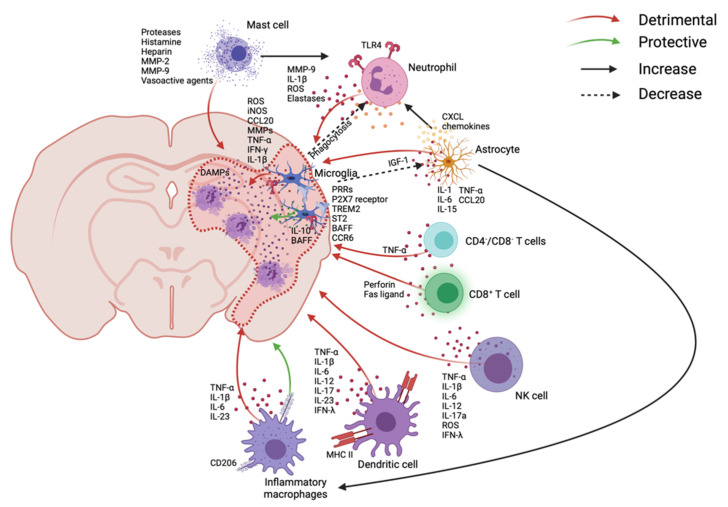
Schematic diagram of the immune responses immediately (<24 h) after stroke onset (created with BioRender.com).

**Figure 2 ijms-22-09486-f002:**
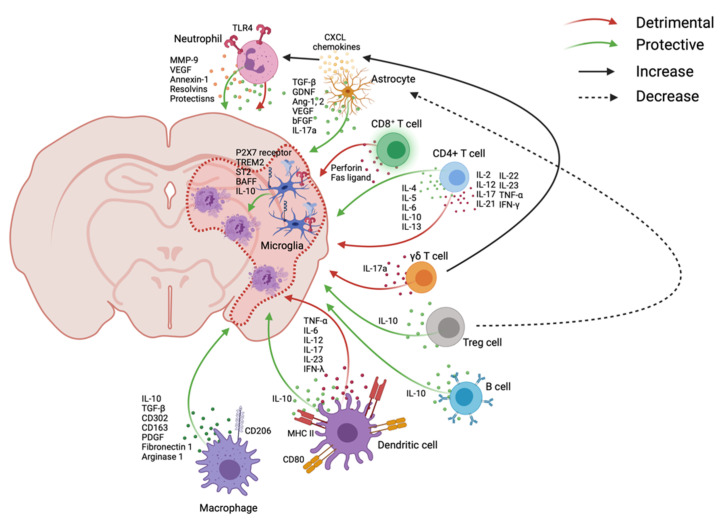
Schematic diagram of the immune responses in the later phase after stroke onset (created with BioRender.com).

**Figure 3 ijms-22-09486-f003:**
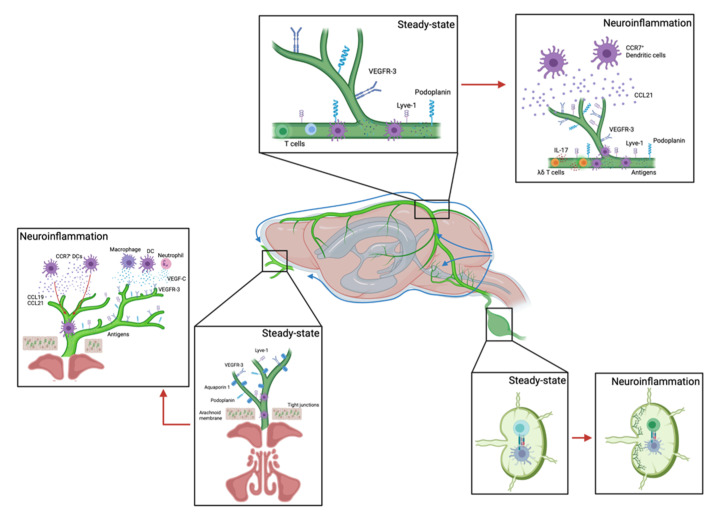
Schematic diagram of lymphangiogenesis occurring in the CNS after neuroinflammation (created with BioRender.com).

**Table 1 ijms-22-09486-t001:** A table summarizing the temporal trend, the produced cytokines/chemokines, and the actions of each immune cell after stroke.

Immune Cells	Temporal Trend	Produced Cytokines/Chemokines	Pro- or Anti-Inflammatory	Action(s)
**Microglia**	Peak at 48–72 h and last for weeks	iNOS	Pro	BBB destruction, leukocyte recruitment, increase adhesion molecules
		MMPs	Pro	
		TNF-α	Pro	
		IL-1β	Pro	
		ROS	Pro	
		RNS	Pro	
		BAFF	Anti	IFN-γ and IL-10 production
		IGF-1	Anti	Suppress astrocyte activities
**Astrocytes**	Shown within 3–5 days	CCL20	Pro	Recruit macrophages
		CXCL chemokines	Pro	Recruit neutrophils
		Adhesion molecules	Pro	Leukocyte recruitment
		IL-15	Pro	Recruit CD8^+^ T-cells and NK cells
		TNF-α, IL-1, 6	Pro	Augment immune responses
		TGF-β	Anti	Develop blood vessels, maintain BBB integrity and endothelial progenitor cells
		GDNF	Anti	
		Angiopoietin 1, 2	Anti	
		VEGF	Anti	
		bFGF	Anti	
		IL-17a	Anti	Brain tissue repair and recovery in later period
**Neutrophils**	Accumulate after 3 h, peak at day 1–3, and dissipate over 7 days	Elastases	Pro	Cerebral edema, BBB destruction, and neuronal death
		MMP-9	Pro	
		IL-1β	Pro	
		ROS	Pro	
		MMP-9	Anti	Degradation of DAMP signaling and vascular remodeling
		VEGF	Anti	Cerebral angiogenesis
		Annexin-1	Anti	Microglia migration toward the infarct core after 1 day
		Resolvins	Anti	Decrease neutrophil migration and pro-inflammatory cytokine release
		Protectins	Anti	
**Mast cells**	Significant increase after 24 h	Histamine	Pro	Destruct BBB, increase vascular permeability, leukocyte recruitment, cerebral edema, destroy tight junctions, and disrupt hemostasis
		Heparin	Pro	
		Vasoactive agents	Pro	
		Chymase	Pro	
		MMP-2, 9	Pro	
**Monocyte**/ **Macrophage**	Shown as early as 3 h, peak at day 3, and become anti-inflammatory at day 7	TNF-α	Pro	Augment immune responses
		IL-1β	Pro	
		IL-23	Pro	IL-17a production from γδ T-cells
		IL-10	Anti	Tissue repair and wound healing
		TGF-β	Anti	
		CD302, 163, 206	Anti	
		PDGF	Anti	
		Fibronectin 1	Anti	
		Arginase 1	Anti	
**DCs**	Detected as early as 1 h and stay elevated till day 7	IL-23	Pro	IL-17a production from γδ T-cells and neutrophil infiltration
		IL-6, 12, 17, 1β	Pro	Augment immune responses
		TNF-α	Pro	
		IFN-γ	Pro	
		IL-10	Anti	Immunosuppression
**NK cells**	3 h, peak at 12 h, and remain elevated at least 4 days	IFN-γ	Pro	Augment immune responses and development of cerebral infarction
		IL-17a, 6, 12, 1β	Pro	
		TNF-α	Pro	
		ROS	Pro	
**CD4-/CD8- T-cells**	1–3 days	TNF-α	Pro	Augment immune responses
**CD8^+^ T-cells**	Detected as early as 3 h and stay for about 30 days	Perforin	Pro	Neurotoxicity and augment immune responses
		Fas ligand	Pro	
**CD4^+^ T-cells (Th1 and Th17)**	Shown at 24 h and stay for about 30 days	IL-2, 12, 17, 21, 22, 23	Pro	Augment immune responses
		TNF-α	Pro	
		IFN-γ	Pro	
**CD4^+^ T-cells (Th2)**		IL-4, 5, 6, 10, 13	Anti	Immunosuppression
**γδ T-cells**	Within 3 days	IL-17a	Pro	Neutrophil infiltration
**Tregs**	Shown after several days and stays for about 30 days	IL-10	Anti	Suppress astrogliosis, regulate astrocyte neurotoxicity, and functional recovery
		IL-17 (in certain conditions)	Anti	Inhibit CD4^+^ T-cell proliferation
**B-cells**	Delayed appearance after weeks of onset	IL-10	Anti	Neuroprotection

## Data Availability

Not applicable.

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
