# Peer review of "Molecular Mechanisms of Neuroimmune Crosstalk in the Pathogenesis of Stroke"

_ijms, 2021, doi:10.3390/ijms22179486_

Round 1
Reviewer 1 Report
Dear Editor,
The manuscript by Choi et al. reviews the roles of immune cascade, lymphatic, water and fluid balance and their homeostatic regulation post-stroke within the brain.
The review is comprehensive, informative, timely and up-to-date (in most parts). Authors were successful in providing some well compiled opinions and summaries. The mechanistic figures and tables can be a good starting point for future studies and will be of interest for IJMS readers and beyond.
However, there is a number of major and minor points that would need to be addressed in order to improve the quality of this paper before it can be accepted for publication.
General:
- This review overlooked some essential and up-to-date work regarding the pathophysiology of stroke and recent advances in target validation and future therapies. I have made some suggestions below but authors are encouraged to consider citing updated references throughout the review, whenever possible.
-Define abbreviations whenever they appear first in the manuscript and use them throughout. For example, LECs in line 649.
Major:
-Certain sections are too long and lack structure. For example, in 2.1 Ischemic Stroke, this part can benefit from subheadings according to each cell type.
-Points to be discussed in more details:
1-Ageing has been mentioned in two sections. It will be useful to discuss the emerging concept of “inflammaging” as a new crosstalk immune–metabolic for age-related diseases and how it can be predisposition factor for stroke. References to be included:
https://www.nature.com/articles/s41574-018-0059-4
https://www.ncbi.nlm.nih.gov/pmc/articles/PMC7407516/
2-Lines 235-238 “. However, after ischemic stroke, accumulation of ions and deregulation of signaling pathways overwhelm astrocytes which leads to induction of neuronal cell death with catabolic processes [105, 237 106]”. The role of brain energetic to be discussed. Authors need to mention the astrocyte‐neuron lactate shuttle (ANLS) hypothesis postulated in 1994 (Pellerin and Magistretti 1994). According to this, astrocytes serve as a ‘lactate source’ whereas neurons serve as a ‘lactate sink’. Moreover, the opposition by Bak and colleagues who argued that oxidative metabolism of lactate within neurons only occurs during repolarization (and in the period between depolarizations) rather than during neurotransmission activity. The emerging role of astrocytes has helped in settling this debate in favour for ANLS hypothesis. References to be included:
https://pubmed.ncbi.nlm.nih.gov/31318452/
https://pubmed.ncbi.nlm.nih.gov/19393013/
https://pubmed.ncbi.nlm.nih.gov/7938003/
3- The authors omitting a key study from 2020, demonstrating that the development of edema following injury-induced hypoxia is AQP4 dependent. That study shows that CNS edema is associated with increases both in total aquaporin-4 expression and aquaporin-4 subcellular translocation to the blood-spinal-cord-barrier (BSCB). Pharmacological inhibition of AQP-4 translocation to the BSCB prevents the development of CNS edema and promotes functional recovery in injured rats.
This role has been recently been confirmed by the work of Sylvain et al BBA 2021 which has demonstrated that targeting AQP4 effectively reduces cerebral edema during the early acute phase in in stroke using photothrombotic stroke model. They have also shown a link to brain energy metabolism as indicated by the increase of glycogen levels. Reference to be included:
https://www.cell.com/cell/fulltext/S0092-8674(20)30330-5.
https://pubmed.ncbi.nlm.nih.gov/33561476/
The authors should review their data in light of this important publication:
4- Section 4: Edema is a hallmark of stroke and AQPs are known to play the main role in brain water homeostasis in health and disease. Three examples of cerebral AQPs have been mention but there was no proper introduction to AQPs. A brief intro should cover some general background about AQP including the following points:
- AQPs are historically known to be passive transporters of water. Lines of evidence in the last decade have highlighted the diverse function of AQPs beyond water homeostasis. Authors need to cover this point. A reference to be included:
https://www.ncbi.nlm.nih.gov/pubmed/26365508
Moreover, a subgroup of AQP water channels also facilitates transmembrane diffusion of small, polar solutes not only water; aquaglyceroporin. References to be included:
https://www.ncbi.nlm.nih.gov/pubmed/16715408
https://www.ncbi.nlm.nih.gov/pubmed/31889130
- The increased AQP4 expression and the redistribution/surface localization can be two different concepts. Previous studies have shown an increased in AQP4 membrane localisation in primary human astrocytes which wasn’t accompanied by a change in AQP4 protein expression levels. This mislocalization can be a potential therapeutic target. This addresses the point in lines 597-599 “Astrogliosis causes dispersion of AQP4 in the cytoplasm of brain parenchyma[250, 597 251], leading to fluid accumulation in the brain parenchyma, tissue swelling (cerebral 598 edema), cell death, and increased intracranial pressure[252, 253].”. References to be included:
https://www.ncbi.nlm.nih.gov/pmc/articles/PMC5765450/
https://www.ncbi.nlm.nih.gov/pubmed/31242419
- AQPs have been validated as an important drug target but there is no single drug that has yet been approved to successfully target it. This needs to be mentioned especially since there aren’t many studies which investigated the mentioned future therapies in term of the communication between AQPs (mainly AQP4) and other therapeutic targets since it’s hard to target stroke using one line of therapies. References to be included:
https://www.ncbi.nlm.nih.gov/pmc/articles/PMC4067137/
https://www.ncbi.nlm.nih.gov/pmc/articles/PMC6480248/
- Lines 634-637: “Another suggested pathway of CSF is using the extracellular spaces in capillaries and arteries to exit the brain in the opposite direction to the arterial blood flow[268]. Since it is not clear what are the roles and functions of the glymphatic system, more research would be necessary for both healthy and disease states during CNS neuroinflammation.”. The field tends to discuss ‘water’ and ‘fluid’ in a manner that incorrectly suggests their interchangeability. However, it is important to note that in the glymphatic system, the clearance of brain waste occurs through paracellular flow. Classic tracer studies measure this paracellular flow, while the use of H217O captures both paracellular flow and diffusive transcellular exchange of water. Importantly, both are AQP4 dependent — one directly and one indirectly. This needs to be clarified. Reference:
https://pubmed.ncbi.nlm.nih.gov/33857020/
Minor:
- Authors need to briefly discuss future directions following towards the end of their discussion and conclusion. This could include, but not limit to, the use of humanized self-organized models, organoids, 3D cultures and human microvessel-on-a-chip platforms especially those which are amenable for advanced imaging such as TEM and expansion microscopy since they enable real-time monitoring of brain penetration, endothelial activation and leukocyte adherence during stroke and related CNS disorders. This is quite important since for example AQP4 in astrocytes has polarised expression which has proven to be varied between 2D and 3D cultures. References to be included:
https://pubmed.ncbi.nlm.nih.gov/30165870/
https://pubmed.ncbi.nlm.nih.gov/33117784/
https://pubmed.ncbi.nlm.nih.gov/31889243/
-Stroke is yet an incurable disease. Authors are encouraged to point to out to some of the recent advances that can be applied for target validation and lead optimisation such as the use of high-throughput screening as have been nicely reviewed by Aldewachi et al 2021 and Del Palacio et al 2016 as they can provide a novel insight that can support the findings new treatments in the future. References:
https://pubmed.ncbi.nlm.nih.gov/33672148/
https://pubmed.ncbi.nlm.nih.gov/26962874/
Best.
Author Response
Reviewer #1:
Dear Editor,
The manuscript by Choi et al. reviews the roles of immune cascade, lymphatic, water and fluid balance and their homeostatic regulation post-stroke within the brain.
The review is comprehensive, informative, timely and up-to-date (in most parts). Authors were successful in providing some well compiled opinions and summaries. The mechanistic figures and tables can be a good starting point for future studies and will be of interest for IJMS readers and beyond.
However, there is a number of major and minor points that would need to be addressed in order to improve the quality of this paper before it can be accepted for publication.
General:
- This review overlooked some essential and up-to-date work regarding the pathophysiology of stroke and recent advances in target validation and future therapies. I have made some suggestions below but authors are encouraged to consider citing updated references throughout the review, whenever possible.
We thank the reviewer for this comment. We tried to update the references and answer the suggestions below.
-Define abbreviations whenever they appear first in the manuscript and use them throughout. For example, LECs in line 649.
We agree with the reviewer and the abbreviations are defined on the modified review. When those abbreviations were used on the title of sections, those abbreviations are defined again as necessary. Please see the attachment for the modified review.
Major:
-Certain sections are too long and lack structure. For example, in 2.1 Ischemic Stroke, this part can benefit from subheadings according to each cell type.
We agree with the reviewer on this comment. We added the subheadings in a few sections on the modified review. Please see the attachment for the change.
-Points to be discussed in more details:
1-Ageing has been mentioned in two sections. It will be useful to discuss the emerging concept of “inflammaging” as a new crosstalk immune–metabolic for age-related diseases and how it can be predisposition factor for stroke. References to be included:
https://www.nature.com/articles/s41574-018-0059-4
https://www.ncbi.nlm.nih.gov/pmc/articles/PMC7407516/
We agree with the reviewer. We combined both contents and created a new section “6. Aging” on the modified review. Inflammaging topic is also added as a subsection under the “Aging” section. These section titles are highlighted in yellow to be easily found on the modified review. Please see the attachment for the change.
2-Lines 235-238 “. However, after ischemic stroke, accumulation of ions and deregulation of signaling pathways overwhelm astrocytes which leads to induction of neuronal cell death with catabolic processes [105, 237 106]”. The role of brain energetic to be discussed. Authors need to mention the astrocyte‐neuron lactate shuttle (ANLS) hypothesis postulated in 1994 (Pellerin and Magistretti 1994). According to this, astrocytes serve as a ‘lactate source’ whereas neurons serve as a ‘lactate sink’. Moreover, the opposition by Bak and colleagues who argued that oxidative metabolism of lactate within neurons only occurs during repolarization (and in the period between depolarizations) rather than during neurotransmission activity. The emerging role of astrocytes has helped in settling this debate in favour for ANLS hypothesis. References to be included:
https://pubmed.ncbi.nlm.nih.gov/31318452/
https://pubmed.ncbi.nlm.nih.gov/19393013/
https://pubmed.ncbi.nlm.nih.gov/7938003/
We thank the reviewer for this comment. This content about ANLS is added on the modified review which is highlighted in yellow under the section “3. Astrocytes”. The paragraphs are highlighted in yellow to be easily found on the modified review. Please see the attachment for the change.
3- The authors omitting a key study from 2020, demonstrating that the development of edema following injury-induced hypoxia is AQP4 dependent. That study shows that CNS edema is associated with increases both in total aquaporin-4 expression and aquaporin-4 subcellular translocation to the blood-spinal-cord-barrier (BSCB). Pharmacological inhibition of AQP-4 translocation to the BSCB prevents the development of CNS edema and promotes functional recovery in injured rats.
This role has been recently been confirmed by the work of Sylvain et al BBA 2021 which has demonstrated that targeting AQP4 effectively reduces cerebral edema during the early acute phase in in stroke using photothrombotic stroke model. They have also shown a link to brain energy metabolism as indicated by the increase of glycogen levels. Reference to be included:
https://www.cell.com/cell/fulltext/S0092-8674(20)30330-5.
https://pubmed.ncbi.nlm.nih.gov/33561476/
The authors should review their data in light of this important publication:
We thank the reviewer for this comment. We added AQP4 and suggested studies under the “3.2 AQPs in Stroke” section on the modified review. This section title and paragraph are highlighted in yellow to be easily found on the modified review. Please see the attachment for the change.
4- Section 4: Edema is a hallmark of stroke and AQPs are known to play the main role in brain water homeostasis in health and disease. Three examples of cerebral AQPs have been mention but there was no proper introduction to AQPs. A brief intro should cover some general background about AQP including the following points:
- AQPs are historically known to be passive transporters of water. Lines of evidence in the last decade have highlighted the diverse function of AQPs beyond water homeostasis. Authors need to cover this point. A reference to be included:
https://www.ncbi.nlm.nih.gov/pubmed/26365508
Moreover, a subgroup of AQP water channels also facilitates transmembrane diffusion of small, polar solutes not only water; aquaglyceroporin. References to be included:
https://www.ncbi.nlm.nih.gov/pubmed/16715408
https://www.ncbi.nlm.nih.gov/pubmed/31889130
We agree with the reviewer on this comment. We added introduction paragraph of AQPs and information about aquaglyceroporin under the “3.1 Aquaporins (AQPs)” section on the modified review. This section title and paragraphs are highlighted in yellow to be easily found on the modified review. Please see the attachment for the change.
- The increased AQP4 expression and the redistribution/surface localization can be two different concepts. Previous studies have shown an increased in AQP4 membrane localisation in primary human astrocytes which wasn’t accompanied by a change in AQP4 protein expression levels. This mislocalization can be a potential therapeutic target. This addresses the point in lines 597-599 “Astrogliosis causes dispersion of AQP4 in the cytoplasm of brain parenchyma[250, 597 251], leading to fluid accumulation in the brain parenchyma, tissue swelling (cerebral 598 edema), cell death, and increased intracranial pressure[252, 253].”. References to be included:
https://www.ncbi.nlm.nih.gov/pmc/articles/PMC5765450/
https://www.ncbi.nlm.nih.gov/pubmed/31242419
We thank the reviewer for this comment. We added this information under the “8. Novel Treatments targeting Neuroimmune Communications and Homeostasis in Stroke” section on the modified review. Please see the attachment for the change.
- AQPs have been validated as an important drug target but there is no single drug that has yet been approved to successfully target it. This needs to be mentioned especially since there aren’t many studies which investigated the mentioned future therapies in term of the communication between AQPs (mainly AQP4) and other therapeutic targets since it’s hard to target stroke using one line of therapies. References to be included:
https://www.ncbi.nlm.nih.gov/pmc/articles/PMC4067137/
https://www.ncbi.nlm.nih.gov/pmc/articles/PMC6480248/
We thank the reviewer for this comment. We added a paragraph including this information under the “8. Novel Treatments targeting Neuroimmune Communications and Homeostasis in Stroke” section on the modified review. This paragraph is highlighted in yellow to be easily found on the modified review. Please see the attachment for the change.
- Lines 634-637: “Another suggested pathway of CSF is using the extracellular spaces in capillaries and arteries to exit the brain in the opposite direction to the arterial blood flow[268]. Since it is not clear what are the roles and functions of the glymphatic system, more research would be necessary for both healthy and disease states during CNS neuroinflammation.”. The field tends to discuss ‘water’ and ‘fluid’ in a manner that incorrectly suggests their interchangeability. However, it is important to note that in the glymphatic system, the clearance of brain waste occurs through paracellular flow. Classic tracer studies measure this paracellular flow, while the use of H217O captures both paracellular flow and diffusive transcellular exchange of water. Importantly, both are AQP4 dependent — one directly and one indirectly. This needs to be clarified. Reference:
https://pubmed.ncbi.nlm.nih.gov/33857020/
We thank the reviewer for this comment. We added the information about this research paper under the “3.2 AQPs in Stroke” section on the modified review. This paragraph is highlighted in yellow to be easily found on the modified review. Please see the attachment for the change.
Minor:
- Authors need to briefly discuss future directions following towards the end of their discussion and conclusion. This could include, but not limit to, the use of humanized self-organized models, organoids, 3D cultures and human microvessel-on-a-chip platforms especially those which are amenable for advanced imaging such as TEM and expansion microscopy since they enable real-time monitoring of brain penetration, endothelial activation and leukocyte adherence during stroke and related CNS disorders. This is quite important since for example AQP4 in astrocytes has polarised expression which has proven to be varied between 2D and 3D cultures. References to be included:
https://pubmed.ncbi.nlm.nih.gov/30165870/
https://pubmed.ncbi.nlm.nih.gov/33117784/
https://pubmed.ncbi.nlm.nih.gov/31889243/
We thank the reviewer for this comment. We added the information in the last paragraph under the “8. Novel Treatments targeting Neuroimmune Communications and Homeostasis in Stroke” section on the modified review. This paragraph is highlighted in yellow to be easily found on the modified review. Please see the attachment for the change.
-Stroke is yet an incurable disease. Authors are encouraged to point to out to some of the recent advances that can be applied for target validation and lead optimisation such as the use of high-throughput screening as have been nicely reviewed by Aldewachi et al 2021 and Del Palacio et al 2016 as they can provide a novel insight that can support the findings new treatments in the future. References:
https://pubmed.ncbi.nlm.nih.gov/33672148/
https://pubmed.ncbi.nlm.nih.gov/26962874/
We thank the reviewer for this comment. We added the information in the last paragraph under the “8. Novel Treatments targeting Neuroimmune Communications and Homeostasis in Stroke” section on the modified review. This paragraph is highlighted in yellow to be easily found on the modified review. Please see the attachment for the change.

Reviewer 2 Report
In the manuscript by Choi et al. the authors summarize recent data and research achievements on molecular mechanisms of the neuroimmune in the pathogenesis of stroke. The review is comprehensive and provides an excellent overview of the currently available literature. I truly enjoyed reading the manuscript and would like to give some suggestions to bring more clarity and improve the overview of the available data. Therefore, I strongly recommend textual revision, otherwise I fear the manuscript is slightly disorganized and difficult to read.
Suggestions:
1. Some parts and paragraphs are difficult to read and should be rewritten.
1) Page 1, line 30-31:"Every organ system requires a mechanism to regulate homeostasis during steady-state conditions and disease."
2) Page 1, line 31-36:"The lymphatic system is considered to be a key regulator of this process by participating in essential processes such as fluid drainage, removal of waste products, and transporting immune cells and lipids." & "Until recently, it was widely accepted that the central nervous system (CNS) is an “immune-privileged” region because it lacks conventional lymphatic vasculature as well as limited peripheral immune cell presence, unlike peripheral organs." There is no link between these two parts.
3) Page 1, line 39-40:"have opened new opportunities to target CNS diseases." Please change to "have opened new approaches to the treatment of CNS diseases."
4) Page 1, line 42-45:"Stroke can result in death and long-term disability with significant complications." & "Currently, there is only one FDA-approved drug, recombinant tissue plasminogen activator (rtPA) for ischemic stroke patients." There is no link between these two parts.
5) Page 2, line 47:"Hemorrhagic patients" Please change to "Hemorrhagic stroke patients"
6) Page 2, line 76:"inhibit reperfusion" please change to "prevent reperfusion"
7) Page 4, line 176-177:"but deficiency of CX3CR1 and CD200 after cell death activates microglia after stroke" please change to "but deficiencies of CX3CR1 and CD200 after cell death activate microglia after stroke."
8) Page 6, line 288-289:"Together this indicates that when the CCL2-CCR2 mediated entry of MDMs to the injury sites is inhibited, stroke outcome worsens." please change to "It indicates that when the CCL2-CCR2 mediated entry of MDMs to the injury sites is inhibited, stroke outcome worsens."
9) Page 7, line323-325:"Interestingly, MHC II+, OX62+ DCs were found in dura mater, leptomeninges, and choroid plexus in rats where there are more opportunities to encounter CNS antigens." please change to "On the contrary, OX62+ DCs were found in dura mater, leptomeninges, and choroid plexus in rats where there are more opportunities to encounter CNS antigens."
10) Page 13, line 511-512:"Moreover, genetic complement and molecular signaling pathways can affect the differences between sex." Can't understand what differences between sex. And it should be "between males and females."
11) Page 13, line 516-517:"male neurons" & "female neurons" please change to "neurons in male" & "neurons in female"
12) Page 14, line 528:"older female animals: please change to "aged female animals"
13) Page 14, line 558:"tau and proteins" please change to "tau proteins"
14) Page 14, line 559-560:"However, endogenous DNA repair functions are decreased with aging." please change to "However, endogenous DNA repair function is decreased with aging."
15) Page 14, line 561:"mitochondrion" please change to "mitochondria"
16) Page 15, line 595-596:"After the stroke, slower fluid drainage and waste removal have been shown which is possibly due to reactive astrogliosis induced by cytokines released from necrotic cells." please change to "After the stroke, slower fluid drainage and waste removal have been shown, which is possibly due to reactive astrogliosis induced by cytokines released from necrotic cells."
17) Page 15, line 600:"after ICP is resolved" please change to "after resolvation of ICP"
18) Page 16, line 677-678:"Highly branched lymphatic capillaries have thin basement membranes and oak leaf-shaped endothelial cells154." What is that 154? Is it a citation?
19) Page 19, line 786: There are 2 spaces between"...spaces in subarachnoid spaces[271]." & "Additionally, AQP1..." Please delete one space.
20) Page 22, line 921:"T and NK cells" please change to "T cells and NK cells"
2. Some citations are missing.
1) Page 1, line 41-42:"Stroke is a primary example of neuropathology that has significant homeostatic disruption, immune cell infiltration, and an imminent need for novel therapies." Citations are required.
2) Page 2, line 56-57:"Ischemic stroke occurs more commonly (87% of stroke patients) due to an occlusion of blood vessels." Citations are required. (Please cite Shekhar et al. International Journal of Molecular Sciences. 2021; 22(4):2074. https://doi.org/10.3390/ijms22042074 ; Tran-Dinh et al. Int J Mol Sci. 2020 Dec 24;22(1):106. doi: 10.3390/ijms22010106. ; Wu et al. Brain Res. 2017 Sep 15;1671:18-25. doi: 10.1016/j.brainres.2017.06.029.)
3) Page 5, line 226-228:"Similarly, neuroprotectin D1 (NPD1), which can also be produced by neutrophils, elicits similar protective effects when administered i.v. in the rat MCAO model leading to a reduction in infarct volume" NPD1 elicits protective effects through TRPC6 pathway. Another review published in IJMS also showed the TRPC6 pathway plays an important role in ischemic stroke. (Please cite Shekhar et al. International Journal of Molecular Sciences. 2021; 22(4):2074. https://doi.org/10.3390/ijms22042074)
4) Page 13, line 495-496:"Even though males and females have similar anatomical vasculature systems, sex differences can be a critical factor in stroke." Citations or references are required.
5) Page 14, line 561-563:"Dysfunction of the mitochondrion and endoplasmic reticulum (ER) can affect cellular Ca2+ processing capability which can increase in ROS production and neuronal apoptosis." There is another review also showed the importance of dysfunction of the mitochondria leading to increased ROS.(Please see Chen et al. Semin Cancer Biol. 2020 Oct 6:S1044-579X(20)30203-0. doi: 10.1016/j.semcancer.2020.09.012.)
3. The manuscript needs linguistic improvement.
4. Abbreviations are not always explained.
1) Page 8, line 399:"DEREG mice" it is better to give a full name or brief explanation for DEREG
2) Page 15, line 582:"AQP4" it is better to give a full name or brief explanation, same to AQP1 & AQP9
3) Page 15, line 590:"CSF and ISF"
4) Page 15, line 594:"TBI", but later in page 22 line 917 you put the full name "traumatic brain injury (TBI)" Have you double-checked or polished your manuscript before submission?
5. Page 13, line 492:"Among those, this review will briefly discuss how age and sex are critical non-modifiable risk factors of stroke." Please explain the reason or criteria why you choose age and sex but not other risk factors? And please explain the relation between neuroimmune mechanism of stroke and these 2 risk factors in this paragraph, otherwise the is lack of logic to link this paragraph to the whole manuscript and topic.
Additionally: I greatly enjoyed reading the manuscript and it will be a valuable resource for further research. I hope the authors’ understanding if my comments above focus on what I think can/should be improved, without stating often enough how positive I felt about the paper in general. Don’t misinterpret this as a negative attitude - the overall impression is very strongly positive. Hence I express my strong support for publication.
Author Response
Reviewer #2:
In the manuscript by Choi et al. the authors summarize recent data and research achievements on molecular mechanisms of the neuroimmune in the pathogenesis of stroke. The review is comprehensive and provides an excellent overview of the currently available literature. I truly enjoyed reading the manuscript and would like to give some suggestions to bring more clarity and improve the overview of the available data. Therefore, I strongly recommend textual revision, otherwise I fear the manuscript is slightly disorganized and difficult to read.
Suggestions:
- Some parts and paragraphs are difficult to read and should be rewritten.
We thank the reviewer for this comment. We rephrased certain sentences mentioned in the comments below on the modified review. These sentences are highlighted in yellow to be easily found. Also, we changed words and corrected certain phrases listed below on the modified review. Please see the attachment for the change.
1) Page 1, line 30-31:"Every organ system requires a mechanism to regulate homeostasis during steady-state conditions and disease."
2) Page 1, line 31-36:"The lymphatic system is considered to be a key regulator of this process by participating in essential processes such as fluid drainage, removal of waste products, and transporting immune cells and lipids." & "Until recently, it was widely accepted that the central nervous system (CNS) is an “immune-privileged” region because it lacks conventional lymphatic vasculature as well as limited peripheral immune cell presence, unlike peripheral organs." There is no link between these two parts.
3) Page 1, line 39-40:"have opened new opportunities to target CNS diseases." Please change to "have opened new approaches to the treatment of CNS diseases."
4) Page 1, line 42-45:"Stroke can result in death and long-term disability with significant complications." & "Currently, there is only one FDA-approved drug, recombinant tissue plasminogen activator (rtPA) for ischemic stroke patients." There is no link between these two parts.
5) Page 2, line 47:"Hemorrhagic patients" Please change to "Hemorrhagic stroke patients"
6) Page 2, line 76:"inhibit reperfusion" please change to "prevent reperfusion"
7) Page 4, line 176-177:"but deficiency of CX3CR1 and CD200 after cell death activates microglia after stroke" please change to "but deficiencies of CX3CR1 and CD200 after cell death activate microglia after stroke."
8) Page 6, line 288-289:"Together this indicates that when the CCL2-CCR2 mediated entry of MDMs to the injury sites is inhibited, stroke outcome worsens." please change to "It indicates that when the CCL2-CCR2 mediated entry of MDMs to the injury sites is inhibited, stroke outcome worsens."
9) Page 7, line323-325:"Interestingly, MHC II+, OX62+ DCs were found in dura mater, leptomeninges, and choroid plexus in rats where there are more opportunities to encounter CNS antigens." please change to "On the contrary, OX62+ DCs were found in dura mater, leptomeninges, and choroid plexus in rats where there are more opportunities to encounter CNS antigens."
10) Page 13, line 511-512:"Moreover, genetic complement and molecular signaling pathways can affect the differences between sex." Can't understand what differences between sex. And it should be "between males and females."
11) Page 13, line 516-517:"male neurons" & "female neurons" please change to "neurons in male" & "neurons in female"
12) Page 14, line 528:"older female animals: please change to "aged female animals"
13) Page 14, line 558:"tau and proteins" please change to "tau proteins"
14) Page 14, line 559-560:"However, endogenous DNA repair functions are decreased with aging." please change to "However, endogenous DNA repair function is decreased with aging."
15) Page 14, line 561:"mitochondrion" please change to "mitochondria"
16) Page 15, line 595-596:"After the stroke, slower fluid drainage and waste removal have been shown which is possibly due to reactive astrogliosis induced by cytokines released from necrotic cells." please change to "After the stroke, slower fluid drainage and waste removal have been shown, which is possibly due to reactive astrogliosis induced by cytokines released from necrotic cells."
17) Page 15, line 600:"after ICP is resolved" please change to "after resolvation of ICP"
18) Page 16, line 677-678:"Highly branched lymphatic capillaries have thin basement membranes and oak leaf-shaped endothelial cells154." What is that 154? Is it a citation?
Yes, it was supposed to be a citation that was accidentally included in the writing as a text. This has been removed on the modified review.
19) Page 19, line 786: There are 2 spaces between"...spaces in subarachnoid spaces[271]." & "Additionally, AQP1..." Please delete one space.
20) Page 22, line 921:"T and NK cells" please change to "T cells and NK cells"
- Some citations are missing.
We thank the reviewer for this comment. We added citations after those sentences mentioned below on the modified review. Please see the attachment for the change.
1) Page 1, line 41-42:"Stroke is a primary example of neuropathology that has significant homeostatic disruption, immune cell infiltration, and an imminent need for novel therapies." Citations are required.
2) Page 2, line 56-57:"Ischemic stroke occurs more commonly (87% of stroke patients) due to an occlusion of blood vessels." Citations are required. (Please cite Shekhar et al. International Journal of Molecular Sciences. 2021; 22(4):2074. https://doi.org/10.3390/ijms22042074 ; Tran-Dinh et al. Int J Mol Sci. 2020 Dec 24;22(1):106. doi: 10.3390/ijms22010106. ; Wu et al. Brain Res. 2017 Sep 15;1671:18-25. doi: 10.1016/j.brainres.2017.06.029.)
3) Page 5, line 226-228:"Similarly, neuroprotectin D1 (NPD1), which can also be produced by neutrophils, elicits similar protective effects when administered i.v. in the rat MCAO model leading to a reduction in infarct volume" NPD1 elicits protective effects through TRPC6 pathway. Another review published in IJMS also showed the TRPC6 pathway plays an important role in ischemic stroke. (Please cite Shekhar et al. International Journal of Molecular Sciences. 2021; 22(4):2074. https://doi.org/10.3390/ijms22042074)
4) Page 13, line 495-496:"Even though males and females have similar anatomical vasculature systems, sex differences can be a critical factor in stroke." Citations or references are required.
5) Page 14, line 561-563:"Dysfunction of the mitochondrion and endoplasmic reticulum (ER) can affect cellular Ca2+ processing capability which can increase in ROS production and neuronal apoptosis." There is another review also showed the importance of dysfunction of the mitochondria leading to increased ROS.(Please see Chen et al. Semin Cancer Biol. 2020 Oct 6:S1044-579X(20)30203-0. doi: 10.1016/j.semcancer.2020.09.012.)
- The manuscript needs linguistic improvement.
We thank the reviewer for this comment. Even though this review has been proof-read and edited before submission, there must be still some grammatical errors and/or language issues in writing that need to be improved. If necessary, we will contact the editor or MDPI for the language editing service.
- Abbreviations are not always explained.
We agree with the reviewer and the abbreviations are defined on the modified review. When those abbreviations were used on the title of sections, those abbreviations are defined again as necessary. Please see the attachment for the modified review.
1) Page 8, line 399:"DEREG mice" it is better to give a full name or brief explanation for DEREG
2) Page 15, line 582:"AQP4" it is better to give a full name or brief explanation, same to AQP1 & AQP9
3) Page 15, line 590:"CSF and ISF"
4) Page 15, line 594:"TBI", but later in page 22 line 917 you put the full name "traumatic brain injury (TBI)" Have you double-checked or polished your manuscript before submission?
- Page 13, line 492:"Among those, this review will briefly discuss how age and sex are critical non-modifiable risk factors of stroke." Please explain the reason or criteria why you choose age and sex but not other risk factors? And please explain the relation between neuroimmune mechanism of stroke and these 2 risk factors in this paragraph, otherwise the is lack of logic to link this paragraph to the whole manuscript and topic.
We thank with the reviewer for the comment. The paragraphs about Sex and Aging have been updated with new information under the section “6. Aging”. The section title is highlighted in yellow to be easily found on the modified review. Please see the attachment for the change.
Additionally: I greatly enjoyed reading the manuscript and it will be a valuable resource for further research. I hope the authors’ understanding if my comments above focus on what I think can/should be improved, without stating often enough how positive I felt about the paper in general. Don’t misinterpret this as a negative attitude - the overall impression is very strongly positive. Hence I express my strong support for publication.
We sincerely appreciate your comments and feedback.

Round 2
Reviewer 1 Report
Dear Editor,
The authors have successfully addressed the majority of my comments and concerns in order to improve the quality of the manuscript.
I believe that the new subheadings, additional sections and updated references, have contributed to enhancing the clarity of the manuscript, which I can now endorse for publication following a minor edition as below.
Best.
Minor:
-Line 726: TGN-020 is marketed as an aquaporin-4 inhibitor on the basis of Xenopus laevis oocyte swelling assays (which have also identified diverse, structurally-unrelated molecules such as acetazolamide, ethoxzolamide and IMD-0354 as aquaporin inhibitors). The inhibitory action of the majority of these molecules is not reproducible in other assays and many have aquaporin-4-independent effects on brain water transport, confounding the interpretation of in vivo studies. The off-target effects of TGN-020 are unknown. Caution therefore needs to be applied in interpreting any data in which TGN020 has been used. The authors should review this paragraph in light of this. Reference to be added:
https://pubmed.ncbi.nlm.nih.gov/34408336/
This should also be used at the end of sentence in line 725.
Author Response
Dear Editor,
The authors have successfully addressed the majority of my comments and concerns in order to improve the quality of the manuscript.
I believe that the new subheadings, additional sections and updated references, have contributed to enhancing the clarity of the manuscript, which I can now endorse for publication following a minor edition as below.
Best.
Minor:
-Line 726: TGN-020 is marketed as an aquaporin-4 inhibitor on the basis of Xenopus laevis oocyte swelling assays (which have also identified diverse, structurally-unrelated molecules such as acetazolamide, ethoxzolamide and IMD-0354 as aquaporin inhibitors). The inhibitory action of the majority of these molecules is not reproducible in other assays and many have aquaporin-4-independent effects on brain water transport, confounding the interpretation of in vivo studies. The off-target effects of TGN-020 are unknown. Caution therefore needs to be applied in interpreting any data in which TGN020 has been used. The authors should review this paragraph in light of this. Reference to be added:
https://pubmed.ncbi.nlm.nih.gov/34408336/
This should also be used at the end of sentence in line 725.
We appreciate the reviewer for this comment. We added more information about the TGN-020 starting from line 726 and how it has been used in rodent models of ischemic stroke and spinal cord injury. These updated paragraphs are highlighted in yellow to be easily found on the updated review. Please see the attachment for the change.

Reviewer 2 Report
Authors made correction according to my previous suggestions. Strongly recommend for publishing.
Author Response
Authors made correction according to my previous suggestions. Strongly recommend for publishing.
We appreciate the reviewer for this feedback.